

# Bimodality in Ensemble Forecasts of 2-Meter Temperature: Event Aggregation

Cameron Bertossa[1], Peter Hitchcock[1], Arthur DeGaetano[1], and Riwal Plougonven[2]

[1]Dept. Earth and Atmospheric Sciences, Cornell University, Ithaca, New York, USA
[2]LMD-IPSL, Ecole Polytechnique, Institut Polytechnique de Paris, ENS, PSL Research University, Sorbonne Université, CNRS, Paris, France

**Correspondence:** Cameron Bertossa (cdb227@cornell.edu)

**Abstract.** A previous study has shown that a large portion of subseasonal-to-seasonal European Centre for Medium-Range Weather Forecasts (ECMWF) ensemble forecasts for 2-meter temperature exhibit properties of bimodality, in some locations reaching occurrence rates of over 30%. This study introduces novel methodology to help identify 'bimodal events', meteorological events which trigger the development of widespread bimodality in forecasts. Understanding such events not only provides insight into the dynamics of the meteorological phenomena causing bimodal events, but also has drastic implications for the skill of forecasts affected. The methodology that is developed allows one to systematically characterize the spatial and temporal scales of the derived bimodal events, and thus uncover the flow states that lead to them. Three distinct regions that exhibit high occurrence rates of bimodality are studied: one in South America, one in the Southern Ocean and one in the North Atlantic. It is found that each region's bimodality appears to be triggered by synoptic processes interacting with geographically specific processes: In South America bimodality is related to Andes blocking events, in the Southern Ocean bimodality is related to an atmospheric wave interacting with sea ice, and in the North Atlantic bimodality is connected to Rossby wave breaking occurring near the Gulf Stream. This common pattern of large-scale circulation anomalies interacting with local boundary conditions suggests that any deeper dynamical understanding of these events should incorporate such interactions.

## 1 Introduction

The use of ensemble forecasts in order to create a probabilistic viewpoint of the future state of the atmosphere has been an important development in numerical weather prediction (Toth et al., 2003; Leutbecher and Palmer, 2008; Vannitsem et al., 2018). Often, these forecast distributions are treated inherently as Gaussian, where the mean of the ensemble is assumed to be the most likely future state and the variance is the uncertainty associated with this prediction. This assumption, however, is not always appropriate. The present work extends a previous study that demonstrated that bimodality is in fact quite common in extended range forecasts of 2-meter temperature from ECMWF (Bertossa et al., 2021, hereafter B21). While this previous study identified the presence of bimodality in forecasts, it did not analyze its cause. The goal of this paper is to identify flow configurations that lead to bimodal forecasts, as a step toward understanding the processes and conditions under which bimodality can arise. We do so by introducing a novel method to identify such configurations. This method clearly demonstrates that bimodality in these forecasts are linked to a diverse set of spatially and temporally coherent weather events.





The impact of particular atmospheric events on the behavior of ensemble spread is an important topic of study since the error growth of an ensemble is known to vary with the state of the atmosphere (Leutbecher and Palmer, 2008; Palmer, 2000).

A specific type of distribution that may arise from ensemble spread is a bimodal distribution. Bimodal distributions occur when some ensemble members in a probabilistic forecast spread in such a way that they form a distinct group separate from the rest of the ensemble. Depending on the dressing method used, this can result in a corresponding probability density function
(PDF) that contains two modes.

Identifying the presence of bimodality can be important in practice because it changes the interpretation of the forecast relative to a Gaussian assumption. This can strongly affect the forecast probability of crossing 'critical thresholds' for decision makers managing risks, noticeable improvements to the forecast skill through postprocessing methodology, and a better understanding of the potential attraction to different atmospheric states, where the latter may take the form of weather regimes
(B21). However, detecting bimodality requires dressing methods that are costly. Understanding the origins of events that cause bimodality can contribute an indication as to when and where higher-order dressing methods should be used.

In this study, the ensemble forecasts of 2-meter temperature identified as bimodal following B21 are used to identify the presence of associated coherent temporal or spatial flow patterns. The forecasts considered are from the European Centre for Medium-Range Weather Forecasts (ECMWF) Atmospheric Model Ensemble extended forecast (ENS extended) ensemble.
This dataset includes 46-day forecasts of 2-meter temperature, for a 50 member ensemble, initialized every Monday and Thursday (Haiden et al., 2019) from December 3, 2015 until January 28, 2021.

The following questions guide analysis: (1) Is the bimodality identified in forecasts of 2-meter temperature spatially and temporally coherent? If so, what are typical associated length and time scales? (2) Can we associate bimodality with typical flow configurations or weather regimes? (3) Is there evidence that different processes may be leading to bimodality in different
seasons or geographical regions?

To answer these questions, individual bimodal forecasts are grouped together based on how the two modes within each forecast develop, specifically, comparing between forecasts which ensemble members reside in each mode. This allows one to identify coherent patterns of bimodality across forecasts based solely on the evolution of the ensemble spread. Once these coherent forecast groups are found, we quantify their spatial and temporal properties to answer (1). Then, since each grouping
is defined by two coherent modes, one is able to (a) examine the different geophysical characteristics associated within each mode and (b) understand what processes may have led to their development, both of which contribute to (2). Finally, examining these properties as a function of season and location naturally leads to (3).

A limitation of the developed methodology, however, proves to be its computational expensiveness. As a result, the clustering algorithm is restricted in several ways. First, the clustering is performed on a few select regions rather than the entire globe.
Regions are chosen based on high occurrence rates and large average separation between modes as determined by B21. Additionally, regions are chosen to sample varied geographical configurations and are characteristically affected by very different atmospheric phenomena. Secondly, as opposed to all 46 days, the clustering algorithm is only performed for the second and third week of forecast lead times, since these weeks exhibit the highest rates of bimodality occurrence in ECMWF forecasts, as well as the fact that the bimodality is still likely to differ in character than that of the climatological distribution at these





lead times. That being said, these restrictions still allow one to explore the development of coherent bimodal events, without conducting an exhaustive study on all processes resulting in bimodality.

This manuscript is organized as follows: In Sect. 2 the cluster analysis used to classify 'bimodal events' is explained. In Sect. 3 this methodology is applied to three regions of interest, the properties exhibited by each region are examined and physical explanations are hypothesized. Section 4 contains a brief discussion on how this analysis might be applied within the

context of weather regimes. Finally, Sect. 5 concludes.

## 2   Data and Methodology

The developed methodology relies on the fact that atmospheric events are known to have an effect on the spread of an ensemble. Since bimodality is simply a particular type of ensemble spread, this notion may be used to link the properties of coherent bimodal groups (such as spatial extent) to the respective properties of the dynamical events which cause them. However, since

bimodal distributions themselves can take many different forms, an additional specification based on which ensemble members reside in the cold versus warm mode of each forecast is used. Explicitly, once a forecast has been deemed bimodal following the procedure defined in B21, a binary membership vector $\boldsymbol{x} = (x_1, x_2, ..x_{N_E})$ can be defined for each forecast, gridpoint, and lead time, where $N_E$ is the number of members in the forecast. Specifically, if member $i$ of the ensemble is in the cold mode, $x_i = 1$; if it is in the warm mode $x_i = 0$ (see Fig. 1(a) for an example).

Groups of forecasts based on the similarities between membership vectors can then be formed using a clustering routine (Sasirekha and Baby, 2013). The specific clustering routine, the parameters used for this study, and a more exhaustive description of the methodology can be found in the appendix material.

It is important to emphasize that the clustering process depends only on which members belong to which mode for each bimodal forecast, and not in any way on the spatial or temporal coherence of the forecasts identified as bimodal. Thus, those

forecasts that are clustered together are done so entirely as a result of their similar ensemble grouping behavior.

### 2.1   Cluster Definitions

To help measure the coherence of formed clusters, we consider how often a given ensemble member lies in the cold mode of forecasts in a given cluster relative to the total number of forecasts within the cluster. If there are $N_C$ forecasts in a cluster and $\boldsymbol{x}^j$ is the membership vector of the forecast $j$, the fraction of forecasts $f_i$ for which member $i$ is in the cold mode is given by

$$f_i = \frac{1}{N_C} \sum_j^{N_c} x_i^j. \tag{1}$$

We then assign the coherency of a cluster by the number of members which lie in one mode or the other in at least 95% of the forecasts within a cluster. Explicitly, $f_i$ must be lower than 0.05 or greater than 0.95 for a member to be considered 'coherent'. Any number of the 50 members of the ECMWF forecast ensemble may or may not meet this standard.

To provide an example of this measure of coherency, a synthetic example cluster is plotted in Fig. 2. This is an idealized case

where the 'cluster' is simply two points in space over a portion of their forecast leads. Both forecasts are made up of 50 member





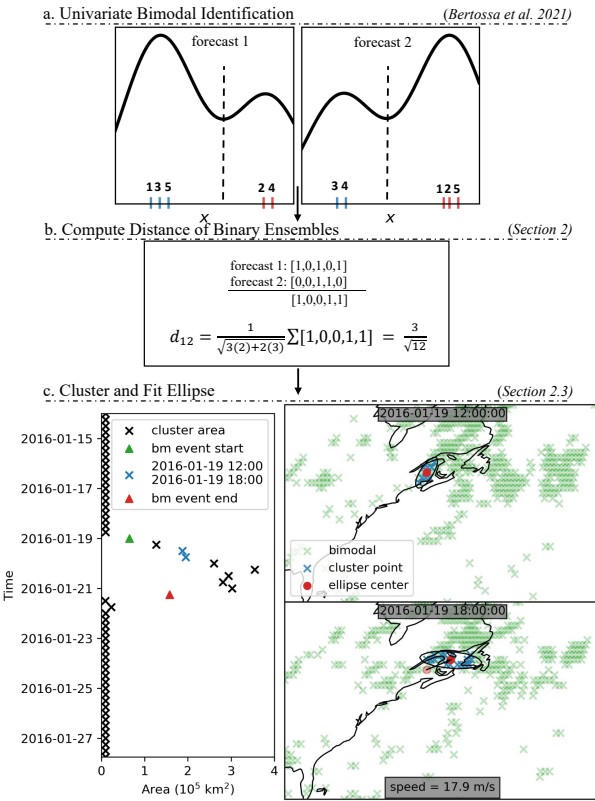

**Figure 1.** Methodology flowchart. Each panel includes a short synopsis and a section label indicating where the process is outlined. (a) Tick labels represent ensemble members. Colors indicate which mode the ensemble member is assigned to based on its position relative to the local minimum of a fit kernel density estimate. (b) Depicts the binary standardization procedure from the forecasts in (a). The distance metric as defined in Section 2 is computed for the two forecasts. (c) Depicts the evolution of a cluster and its fit ellipse. The left panel is the evolution of a cluster's fit ellipse area. The entirety of lead times in which the clustering routine considers is provided (weeks two and three of forecast lead times). The green triangle represents when a bimodal event would start based on the definition presented in Sect. 2.4; the red triangle is when that event would end. Blue x tick marks represent the two particular forecast lead times depicted for the cluster in the right panel. For the right panel, green ticks represent all forecasts which are bimodal at that given lead time. Blue tick marks represent those bimodal forecasts which belong to the same cluster. An example of the ellipse fitting for these two time steps is presented. The black arrow indicates the movement of the ellipse center from one time step to the next and can be used as a means to calculate the discrete speed (listed at the bottom) and direction of propagation.



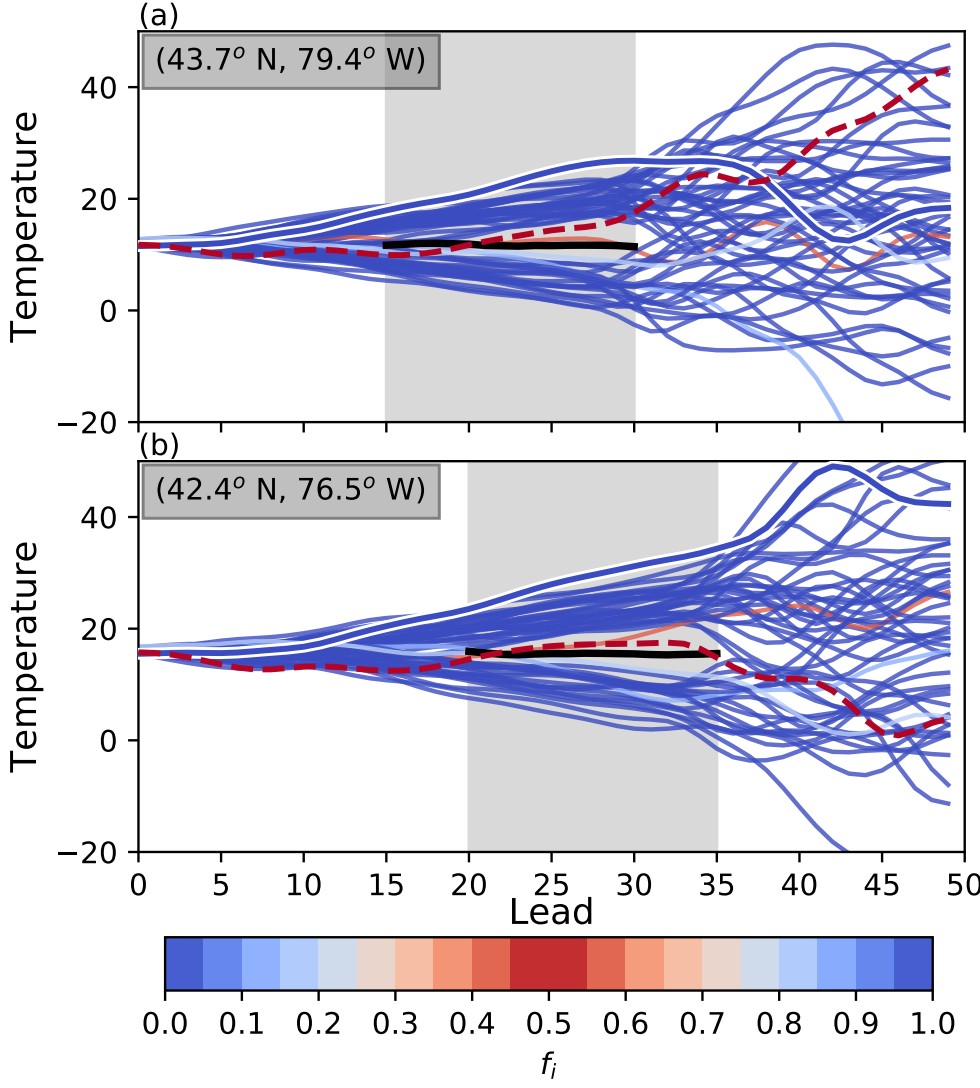

**Figure 2.** A synthetic example to depict coherency. Each panel represents a different forecast ensemble's evolution as a function of lead time. Both forecasts belong to a single cluster which contains only these two spatial points but for the entirety of each forecast's bimodal lead times. Gray shading indicates when an ensemble is bimodal. Thick black lines indicates the fit KDE's relative minima. Dark blue lines indicate ensemble members that have high coherency whereas redder lines indicate low coherency. Correspondingly colored lines in forecast(a) and forecast(b) represent the same ensemble member. Dashed and solid lines are discussed further in the text.

ensembles. The forecast in Fig. 2(a) is bimodal from all leads 15-30 and the forecast in Fig. 2(b) is bimodal from leads 20-35, thus the occupancy of this cluster is equal to 32. In practice, clusters are found to be made up of many different locations. However, this figure offers a more intuitive example of coherency. Dark blue lines are ensemble members that have highly consistent mode occupancy, while red lines indicate low consistency. Correspondingly colored lines in the upper forecast and





lower forecast indicate the same ensemble member. For example, the dashed dark red line in both forecasts represent the same member, which has especially low mode consistency in the cluster. For a portion of the bimodal lead times, this particular member is in the cooler mode of forecast (a) but in the warmer mode of forecast (b). Furthermore, in both forecasts, the ensemble member crosses over the relative minimum and resides in the opposite mode for some time. Both of these factors contribute to its poor mode consistency within the cluster as a whole. The dark blue member with the white outline, in contrast,

represents a member with high mode consistency. It resides in the warmer mode of both forecasts for the entirety of their bimodal periods. This particular cluster has a coherency of 46-members, meaning 46 out of the 50 members have values of $f_i > 0.95$ or $< 0.05$.

Finally, this study differentiates between the occupancy and the size of a cluster. Occupancy refers to the total number of univariate bimodal forecasts at all gridpoints and lead times within the cluster. Size refers to the spatial extent spanned by a

cluster at a single lead time (or an average over a set of lead times). In the particular case for Fig. 2, the occupancy of the cluster is equal to 32, as previously mentioned. In contrast, the size of this cluster may be very small since it only consists of two forecast locations.

## 2.2 Cluster Behavior

The three particular regions that are studied in this manuscript are outlined in Fig. 3(a) and Fig. 3(b). Figure 3(c-f) depicts the

occupancy and coherency of the largest 30 clusters found in each forecast for the three regions' cold and warm season. Only the largest 30 clusters for each forecast are retained for analysis because smaller clusters are found to only contain a few dozen forecasts. For each region, the 30 largest clusters include about 20% of the total bimodal forecast occurrences during weeks two and three of forecast lead times. This relatively small proportion is likely due to a sub-optimal clustering routine and is discussed further later in the study. Even within the largest 30 clusters, relatively small clusters are still quite common, with

many occurrences on the order of a couple hundred forecasts or fewer. To ensure that analysis focused on only large-scale dynamical events, later cluster analysis is further thinned to only those clusters that are greater than the 85th percentile in occupancy for each region and season. Each respective 85th percentile occupancy threshold is demarcated in the legends in Fig. 3(c) and Fig. 3(d). Figure 3(e) and Fig. 3(f) depict the distribution of cluster coherency for the largest 30 clusters of each forecast region. Hatched bars indicates this same metric for only those clusters which qualify the appropriate 85th percentile

occupancy threshold.

From Fig. 3(c-f) it becomes apparent that each region's clusters are generally larger (in occupancy) in the cold season as compared to the typical warm season cluster. Additionally, the South American region generally has the smallest clusters while the Southern Ocean region has the largest clusters. However, the South American region exhibits the greatest coherency, while the Southern Ocean region exhibits the least. Cluster coherency is found to be generally worse for clusters with larger

occupancy than clusters with lower occupancy; likely tied to this property, all three regions exhibit greater coherency in their respective warm seasons as compared to their cold seasons. That being said, even when considering clusters larger than the 85th percentile, the mean coherency is still relatively high, as most of the hatched coherency distributions lay above 25 members.



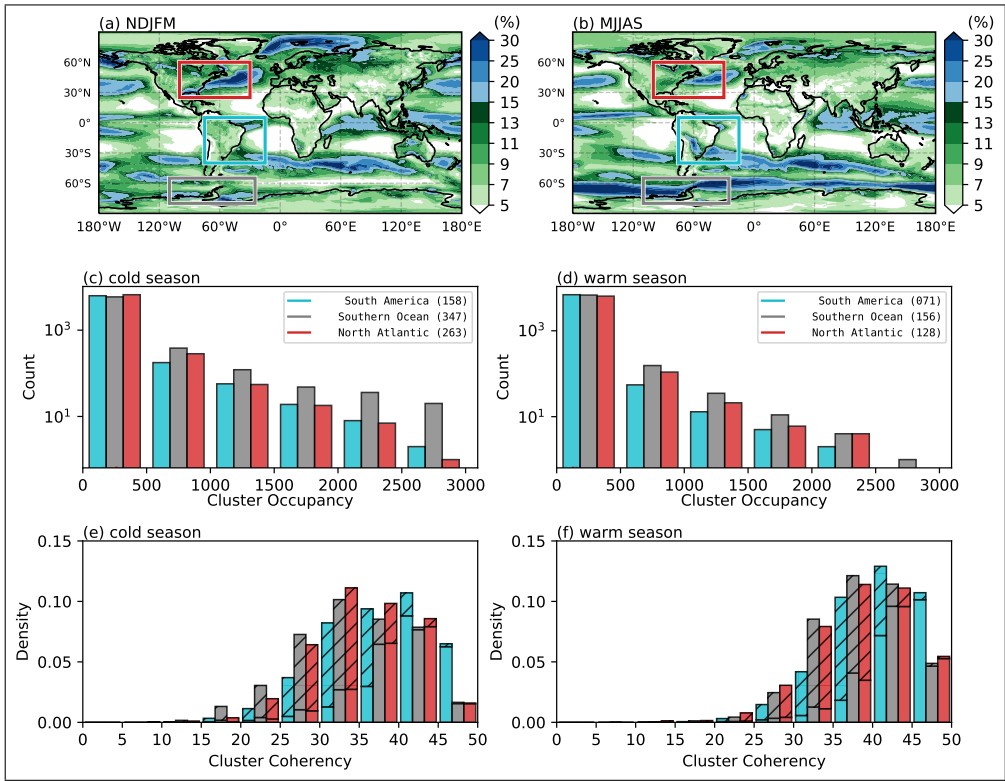

**Figure 3.** (a) From (Bertossa et al., 2021), the occurrence of bimodal forecasts in ECMWF 50-member 2-meter temperature forecasts for November, December, January, February and March (NDJFM). (b) As with (a) but for March, June, July, August and September (MJJAS). (c) The distribution of occupancy for the largest 30 clusters generated from each ECMWF forecast from the cold season of each region (NDJFM for the northern hemisphere and MJJAS for the southern hemisphere). Colors align with the boxed regions in (a) and (b). In the legend, the 85th percentile in occupancy for each region during that particular season is listed in parentheses. (e) Stacked histogram of the number of members in each ensemble with at least 95% coherency ($f_i > 0.95$ or $< 0.05$) for the clusters generated from the cold season of each region. Hatched bars depict the same metric but only for those clusters which are above the 85th percentile in occupancy. (d) as with (c) but for the warm season of each region (MJJAS for the northern hemisphere and NDJFM for the southern hemisphere). (f) as with (e) but for the warm season of each region.

Further analysis is limited to only those forecasts which surpass the 85th percentile in occupancy of their appropriate season and region.

The spatial and temporal properties of the clusters that form may be assessed by plotting how the average cluster changes as a function of lead time. Since, however, the leads in which one cluster exists has no relation to the leads of another cluster, this must be standardized in some way. To do so, the lead time in which a cluster has its greatest occupancy is found and this is the normalization factor for the lead times preceding and following this maximum. The result of this analysis for each region is plotted in Fig. 4(a) and Fig. 4(b). Note that some of the clusters may reach a maximum size very near the beginning of week 2



or very near the end of week 3, the bounds of the cluster analysis. Thus in order to avoid inaccurate representations of evolution near these boundaries, the forecasts whose maxima occur within the first 2.5 days of week 2 or last 2.5 days of week 3 are not included in this figure (roughly 30% of the cases are removed).

An interesting property that emerges for all three regions is that clusters, on average, monotonically increase in occupancy, reach a peak, and then monotonically decrease as a function of lead time. This temporal dependence is in no way forced when

defining these clusters, this property emerges naturally from the analysis. This suggests that these clusters may be capturing the complete life cycle of the events which cause the bimodality to develop. Another interesting property is that the relative occupancy evolutions do not appear to vary greatly between the three regions. That being said, note that this is a depiction of the average cluster structure and particular clusters may vary to a greater extent.

Figure 4(c) and Fig. 4(d) depict how the cluster coherency varies as a function of lead time. As with Fig. 4(a) and Fig. 4(b),

this is plotted in relation to the lead with the maximum occupancy. Similar to Fig. 4(a) and Fig. 4(b), the coherency appears to steadily grow, reach a peak, and then decrease surrounding the lead time in which the occupancy is greatest for a given cluster. This stands in contrast to the relationship found in Fig. 3, in which coherency generally decreases as cluster occupancy increases. Also in contrast to Fig. 4, the magnitude of the coherency varies little depending on the region and season of the cluster (with even some decreased coherency during the warm season for negative lags in the South American region). This

reinforces the suggestion that these clusters are developing around true dynamical events that express themselves explicitly and consistently throughout the groups of forecast distributions.

While so far the properties related to the number of points within a cluster as it evolves have been evaluated, one may also wish to understand the spatial structure of these clusters. Are those points which comprise the cluster randomly distributed throughout the region's domain or are they concentrated together in space? The latter is found to be the far more common

occurrence, with one example being that from Fig. 1(c). This is examined quantitatively by first fitting a 'box' around clusters for each lead time; where the upper bound of the box extends to the northern most cluster point, the lower bound extends to the southern most, the left bound extends to the western most and the right bound extends to the eastern most. Then, the relative fraction of gridpoints within that bounding box which are a part of the cluster is found, high spatial coherence will be accompanied by a larger fraction. Since this procedure requires some number of points to reside within a cluster at a particular

lead time for it to be performed, an arbitrary threshold of at least one-tenth the maximum occupancy (>0.1 for the y-axis in Fig.4(a)) is chosen. It is found that for these lead times, 40-50% of the bounding 'box' is filled with points belonging to the cluster for all three regions, with slightly higher percentages occurring closer to the maximum relative occupancy for each cluster. This leads one to believe that the clusters exhibit relatively high spatial coherence, especially since the bounding box has been defined very conservatively (lending itself to a lot of erroneous area).

In order to better generalize the spatial extent of a cluster, an ellipse is fit to the points within the cluster at a given lead time. Explicitly, this is done using a covariance error ellipse, where 90% of the cluster points at a given lead time are contained within the confidence ellipse. The advantage of using an ellipse rather than the 'box' as described above is that it offers more flexibility if the clusters are oriented diagonally across the domain (erroneous area will not be counted as drastically). The area of this ellipse is then used as the representative area of the cluster at that lead time.





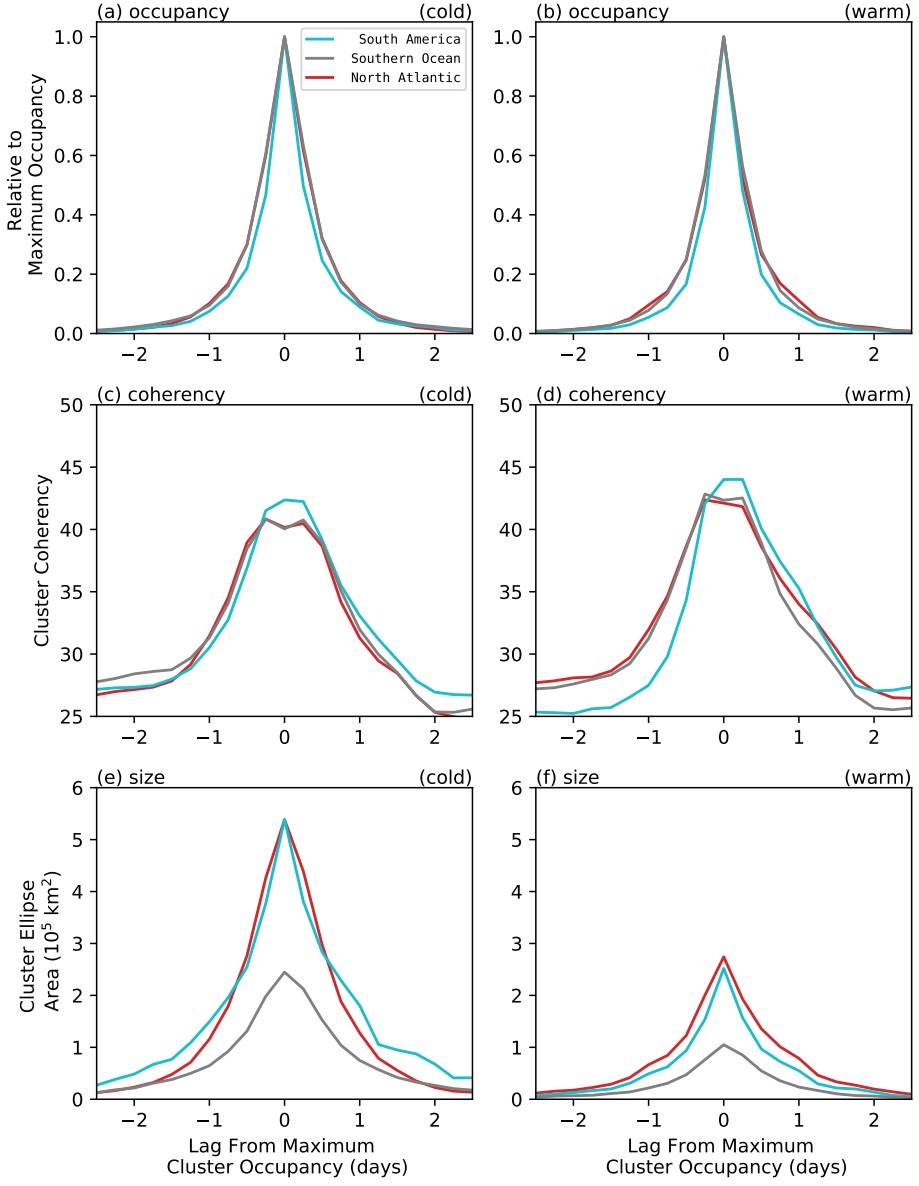

**Figure 4.** (a) The mean cluster occupancy size as a function of lead time for each region's cold season. Cluster sizes are normalized based on the maximum cluster size at a single lead time. Lags indicate the lead time preceding or following when the maximum cluster occupancy occurred. (c) As in (a) except the y-axis is the cluster coherency at each lead time. (e) As in (a) except the y-axis is the cluster area according to a fit covariance error ellipse. (b), (d), (f) as with (a), (c), (e) but for each region's warm season. Each plot only includes the clusters that exceed the appropriate 85th percentile in size. Additionally, some of the clusters may reach a maximum size very near the time bounds of our cluster analysis. Thus in order to avoid inaccurate representations of evolution near these boundaries, the forecasts whose occupancy maxima occurred within the first 2.5 days or last 2.5 days of our analysis were not included in this figure (roughly 30% of the cases were removed).





Fig. 4(e) and Fig. 4(f) depict how the cluster area varies relative to the maximum cluster occupancy. Similar to Fig. 4(a) and Fig. 4(b), the area of the typical event appears to steadily grow, reach a peak and then decrease. The North Atlantic and South American regions have similarly sized events whereas the Southern Ocean's events are typically smaller. There may, however, be some amount of bias due to the effect of latitude on gridbox size. All three regions have events that are on the order of $10^5$ km$^2$ for approximately 2 days, with larger events occurring during the cold season on average as compared to the warm season.

The combination of Fig. 4(a-f) supports the coherent spatial and temporal structures that are resolved with this clustering methodology (a primary goal of this study), without any forcing by the algorithm. Clusters not only appear to capture the maturing and deformation of an event, but also have coherency, where well over half of the ensemble members are consistently in one mode but not the other for 95% of the points within the cluster.

## 2.3 Defining a 'bimodal event'

Supported by the information exhibited in Fig. 4 and by the spatial coherence exhibited through the bounding 'box' procedure, we choose to define 'bimodal events' by only the leads in which the cluster occupancy continuously exceeds 10% of its maximum value. If a cluster exceeds this threshold, dips below the threshold and then re-exceeds the threshold, the longest continuous stretch of lead times is used. This allows one to define an event's persistence, or length, based on the longest continuous length of lead times in which this threshold is satisfied. This more restrictive definition reduces noise present in the

cluster and contributes to increased coherency when examining a bimodal event. Additionally, the use of the fit ellipse may be more effective (and thus the area derived from it) since it's not representative of only a few points. Refer to the left panel of Fig. 1(c) for an example of the size evolution of a fit ellipse for a single cluster.

Other characteristics of these bimodal events can be examined through the properties of the fit ellipse. Namely, to understand the direction of movement and speed of these events, the behavior of the ellipse itself can be used. This study chooses to use

the velocity of the ellipse center from one lead time to the next as a means of studying the bimodal event velocity. An example of this process is depicted in the right panel of Fig. 1(c) for a set of timesteps. In this case, the ellipse, and thus the event, propagates eastward with speed of roughly 18 ms$^{-1}$.

Using only the forecasts which qualify as 'bimodal events' comprises 10% of the total bimodal occurrences for weeks two and three of forecast lead times found in Bertossa et al. (2021). It is clear, at least with how the clustering process has

been defined for this study, that many of the total bimodal forecasts are not being used. Part of this (likely a large portion) is a function of the clustering algorithm and the suboptimal parameters chosen for it; much of the 'signal' of these bimodal forecasts has been thrown out along the way by the choices made in this study. See, for example, the larger extent of bimodality that appears to be spatially coherent, yet is not included within the cluster in Fig. 1(c). Though unnecessary for answering the guiding questions presented here (as will be shown), the clustering procedure may need to be more explicitly optimized in the

future to take full advantage of the bimodal forecast set.



## 3 Regional Analysis

This study thus far has created a definition for bimodal events and presented tools (ellipse fitting) and metrics (coherency, size, length, velocity) with which these events can be studied and compared. These metrics have been applied generally (Fig. 1(c) and Fig. 4) and have shown some potential in examining events which lead to outbreaks in bimodality. However, we now
explore if the tools which have been developed in this study also exhibit 'usefulness' in a more concrete application: can these tools help explain what meteorological phenomena lead to the development of bimodality. In this section, a combination of ellipse properties, spatial occurrence maps and case studies are used to give plausible explanations for phenomena resulting in bimodal events for each of the three regions of interest. The properties and hypothesized phenomena for each region are first presented (guiding question (1) presented in the introduction), then, Sect. 4 provides linkages between the regions and briefly
discusses how these findings may be applied within the context of regimes (guiding questions (2) and (3)).

### 3.1 South American Region

The analysis of the South American region begins with a short discussion of the distributions of event length, event area, and event velocity based on fit ellipses for the region's bimodal events. These properties are presented for the cold season in Fig. 5(a,c,e), and for the warm season in Fig. 5(b,d,f). In general, events persist for roughly 1-2 days with slightly longer
events occurring on average during the cold season as compared to the warm season. In agreement with Fig. 4, events are on the order of a few $10^5$ km$^2$. However, there is a noticeable size difference between the seasons, with the average cold season event being twice as large as the average warm season event. Events in this region propagate at a speed of roughly 10 ms$^{-1}$ on average, with slightly faster speeds being more common during the cold season. The bimodal events' direction of motion is typically eastward, with less frequent occurrences of NW/SE propagation, which becomes slightly more prevalent during the
cold season.

Next, where these events typically occur within the region is explored. This analysis tracks where events occur based on where the ellipse center, or centroid, lies at a given lead time. Figure 6 depicts a density plot of centroids for all the lead times of defined events. The shading indicates the number of event days per season (e.g. the number of forecast days identified within a bimodal event) within a 5 by 5 degree box. Darker shades indicate more common centroid locations. Note that different
forecast initialization dates may have a bimodal event at the same location for the same validation date, since validation dates can overlap for different forecast initializations. That is to say, for example, 30 event days for an extended season that is 150 days long does not necessarily mean that 20% of the days are predicted to be bimodal events, but rather, of all the forecast lead times analyzed within a given season, 30 days worth, on average, have a bimodal event occurring. Since forecasts are initialized twice per week, 14 days per forecast are considered (weeks two and three), and there are roughly 21 weeks, this
equates to roughly 588 forecast days per extended season. The centroid tracks of fifteen randomly selected bimodal events are also shown to provide some sense of typical trajectories. The location at the start date is indicated by a white triangle, and each cluster centroid proceeds towards the red triangle which indicates the end date. The number of individual tracks plotted is heavily thinned from the total number of bimodal events (hundreds) for clarity.



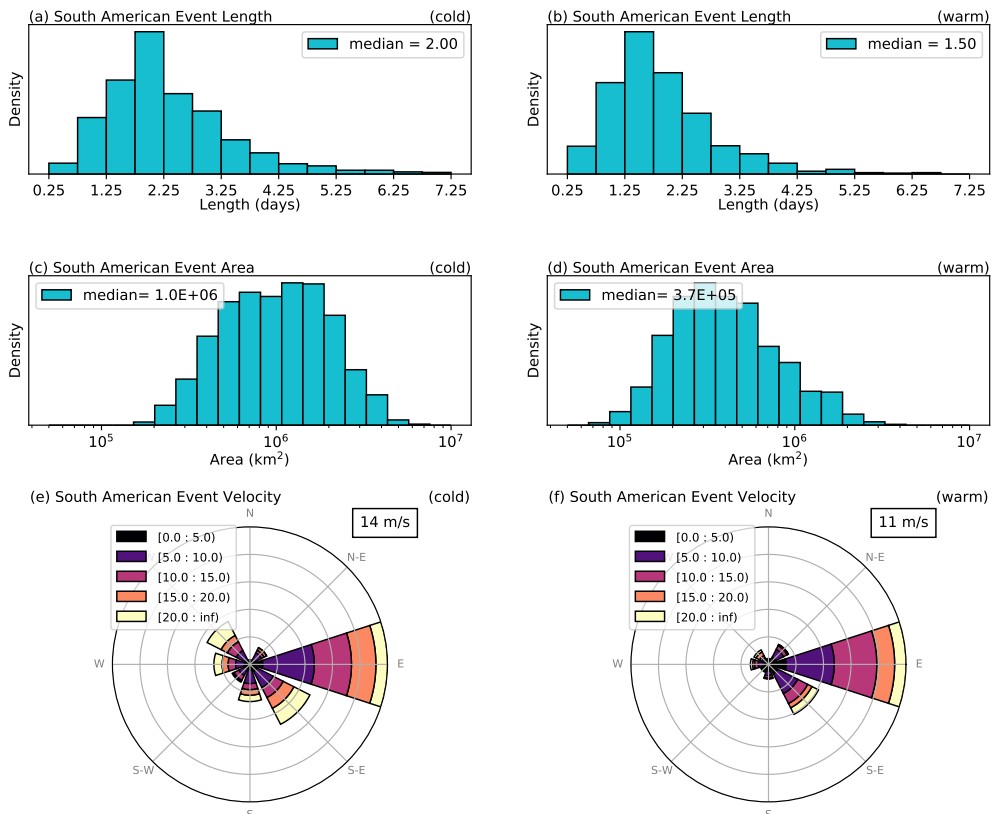

**Figure 5.** (a) Bimodal event length for South America's cold season (May, June, July, August, September). Event length is defined by the longest continuous stretch of cluster lead times that exceed occupancies greater than 10% of the maximum occupancy. (c) Bimodal event area based on the mean of the area of a fit ellipse for each bimodal event. (e) Bimodal event velocity based on the movement of the ellipse center for each bimodal event. Note that a single event's velocity is determined based on the index of the median direction of motion. Height of bar indicates the relative proportion of the direction of propagation for all bimodal events. Bars point towards direction of propagation. Width of color for a given direction indicates the relative proportion of speed for that direction for all bimodal events. The mean speed of all events is listed in the legend. (b,d,f) As with (a,c,e) but for the warm season (November, December, January, February, March).

Figure 6 suggests that bimodal events in warm season commonly occur off the eastern coast of Brazil and Argentina and
propagate eastward across the Atlantic. During the cold season, an additional set of bimodal events are identified around the eastern edge of the Andes, reaching a peak rate of nearly 40 forecast days per season. By analyzing individual tracks it can also be seen that the events near the Andes appear to be relatively stationary, with slight movement south to north. This is distinct compared to the continuous propagation west to east associated with the Atlantic maximum. The alignment along the Andes may explain the increased NW-SE propagation seen in Fig. 5(e).

In order to explore the Andes maximum more thoroughly, a specific bimodal event is analyzed in Figure 7. This forecast is initialized on August 10th 2017, during the region's cold season, and is valid for August 19th 2017. Representative information




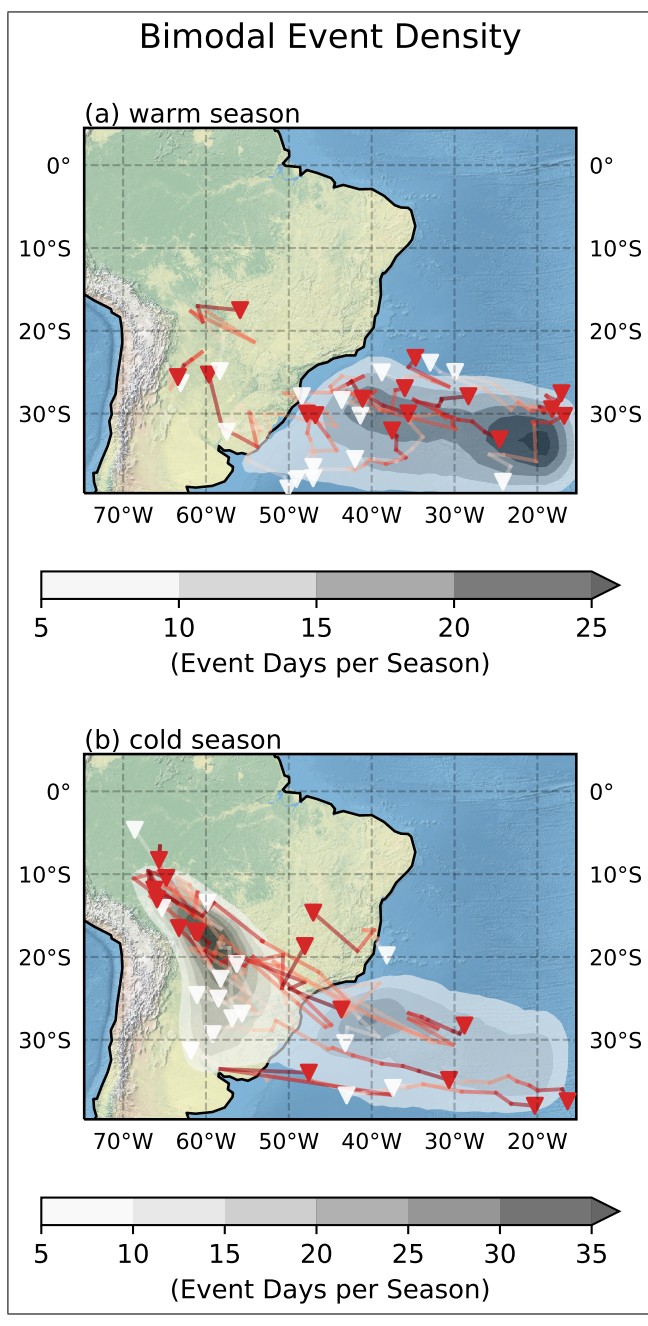

**Figure 6.** Bimodal event density (shaded) and 15 randomly sampled individual track plot (lines and triangles) for the South American region. Individual tracks are depicted by white triangles (start) and red triangles (end), with gradient white to red lines connecting the two. For the density plots, darker shades indicate where events more commonly occur. (a) Forecasts initialized in November, December, January, February and March. (b) Forecasts initialized in May, June, July, August and September.





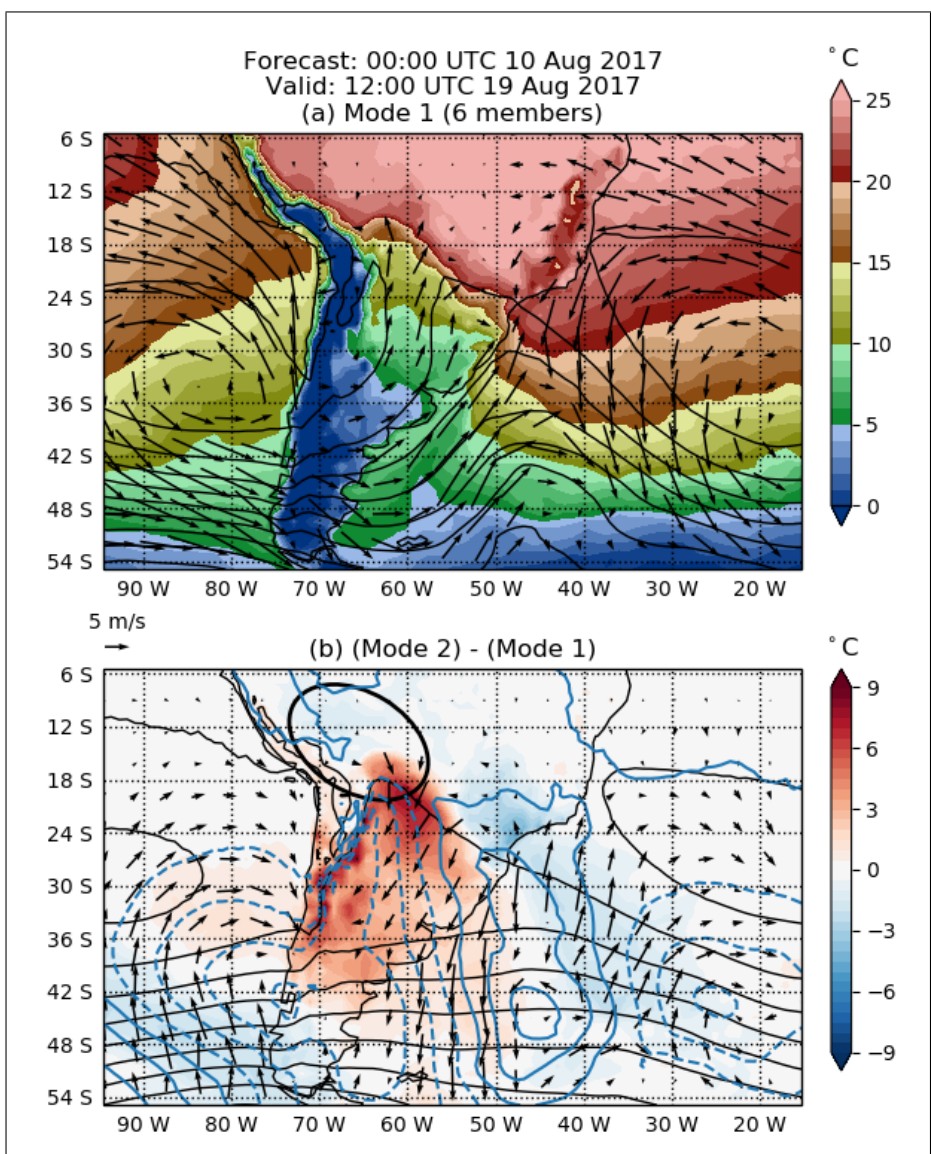

**Figure 7.** A ECMWF forecast initialized on August 10, 2017 and valid for August 19, 2017. (a) Black contours represent the mean 850mb geopotential height contour for the ensemble members assigned to mode 1. Arrows indicate the mean 10m winds for the ensemble members assigned to mode 1. Shaded contours indicate the mean 2-meter temperature of the ensemble members assigned to mode 1. (b) Black contours represent the mean 850mb geopotential height contour for the ensemble members assigned to mode 2. Blue contours represent the difference in the mean 850mb geopotential height from mode 2 and mode 1. Arrows indicate the difference in 10m winds from mode 2 and mode 1. Shaded contours indicate the difference in 2m temperature between mode 2 and mode 1. The black ellipse indicates how the clustering algorithm has fit the event. Black contour intervals are 30 m. Blue contour intervals are 25 m.





from the two modes of the 'medoid' has been plotted. The medoid is defined as the ensemble grouping that is most representative of the cluster. In other words, this is the membership vector (refer back to Fig. 1) that occurs most frequently within the forecasts belonging to this bimodal event's cluster. The properties of each mode of this event are analyzed by taking the mean

of the ensemble members that have been assigned to one mode or the other based on the medoid. This particular event has a coherency of 27 members.

Notable features in the geopotential height field in mode 1 (Fig. 7(a)) include the high pressure systems to the west and east of the continent which are separated by a trough with an axis roughly around 50 $^o$W. The particular location of the western high in Fig. 7(a) causes it to interact with the topography of the Andes mountains, and the eastern edge of the high wraps around the

Andes. The large majority of the accompanying winds to these isobars are geostrophic with some ageostrophic flow towards the northern portion of the event. This brings cold air from the south as can be seen in the temperature field. The result of this southerly flow is reflected in the shaded temperature field, where the center of the continent has anomalously cold temperatures compared to the western and eastern portions.

Figure 7(b) displays a succinct summary of mode 2 (black geopotential lines), as well as the differences between those of

mode 1's geopotential lines (blue contours), 2-meter temperatures (shaded), and 10-meter winds (arrow). Mode 2 (Fig. 7(b)) lacks almost any wave-like behavior in the geopotential heights and isobars remain relatively zonal. The west-to-east high-low-high departure in mode 1 is quite evident in the blue contours in Fig. 7(b). The lack of the wrapping high around the Andes leads to much warmer central South America temperatures in mode 2 because cold air advection from high latitudes is not occurring.

The geopotential height and wind structure exhibited in mode 1 is characteristic of a cold air incursion. This phenomenon is described in detail in Garreaud (2000). These events lead to cold air damming on the eastern edge of the Andes, qualitatively agreeing with the pattern evident in Fig. 7(a). See Garreaud (2000, their Fig. 4b) for a comparison. Furthermore, cold air incursions are associated with 5 $^o$C per day of cooling, quantitatively agreeing with the temperature differences experienced between mode 1 and mode 2 for locations within the influence of this event.

The two modes exhibited in Figure 7 are representative of many of the cases looked at in this region. It would appear that the bimodality that develops in the Andes region is a result of the development in some members of a cold incursion event (like those in mode 1) while others do not predict this phenomenon occurring at that lead time (like those members in mode 2). At least in this case, the clustering methodology that has been presented has led back to a specific dynamical event (and a quite well documented one), exhibiting usefulness, despite the preliminary nature of the algorithm.

**3.2  Southern Ocean Region**

Figure 8 depicts the event persistence, area and velocity for the Southern Ocean region's cold and warm season. Typically, bimodal events persist for roughly 2 days. The median event length is slightly longer for the cold season as compared to the warm season. Additionally, the median event in the cold season is over double the size of the median event in the warm season. Events in the Southern Ocean region propagate at a mean speed of 5-10 ms$^{-1}$, with faster events generally occurring on

average during the cold season. The direction of propagation is mainly west to east in the cold season, with a smaller fraction



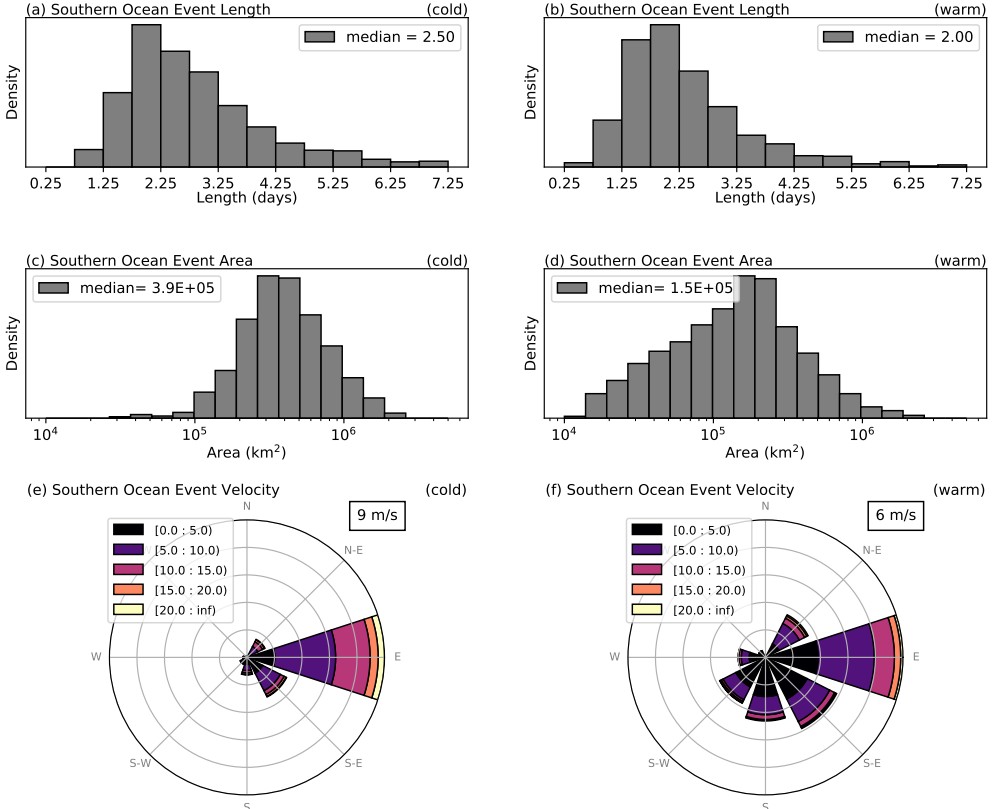

**Figure 8.** As with Figure 5 but for the Southern Ocean region.

of propagation to the southeast. During the warm season, the direction of propagation is much more evenly split, with some southward and south-westward motion as well.

The centroid density and individual tracks for the Southern Ocean's cold and warm season are plotted in Fig. 9. In the cold season, Southern Ocean events mainly propagate west to east above the Antarctic Peninsula. However, during the warm season,
bimodal events are more concentrated near the Ronne Ice Shelf and tracks do not appear to travel as far.

Figure 10 depicts a forecast initialized during the Southern Ocean cold season and valid for July 2, 2019. Mode 1 has a weak low to the west of the Antarctic peninsula. Mode 2, in contrast, has a ridge axis in roughly this same location with a low that is further downstream near 15 $^o$W. This combination results in more meridional flow for mode 2 as compared to mode 1 (as denoted by the large portion of north-south oriented arrows in Fig. 10(b)). Extreme temperature departures between the
modes occur both to the west and to the east of the peninsula, aligning with the departures in low level winds. Mode 2's warm temperature departures (west of the peninsula) are more poleward as compared to the cold departures (east of the peninsula).

Prior to presenting a hypothesis based on this forecast alone, we examine the temperature of each mode for *all* bimodal events within this region. Figure 11(a) depicts the most common temperature for each mode of all bimodal events that occur



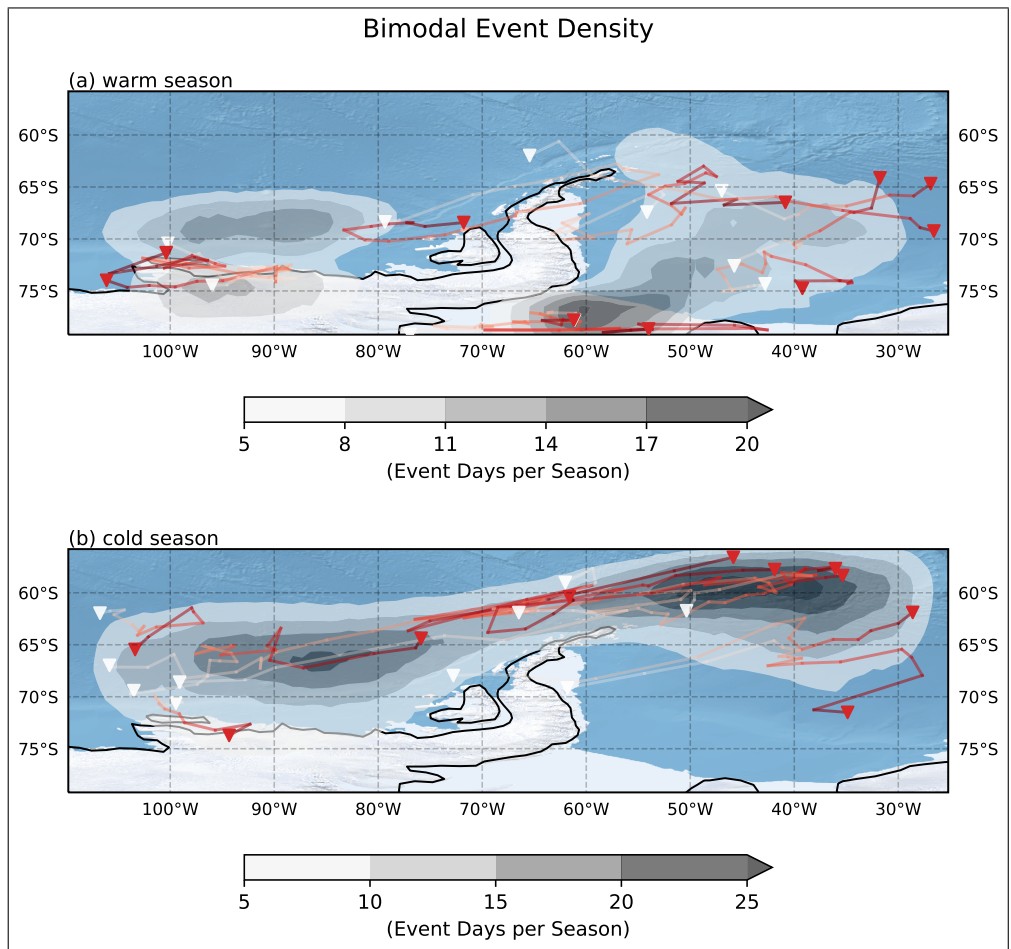

**Figure 9.** As with Figure 6 but for the Southern Ocean region. Note that only 10 randomly selected tracks have been plotted in this case.

in the Southern Ocean region. The temperature of the warmer mode is plotted in red while the cooler mode is plotted in blue.

Accompanied is a PDF of the values of each mode (Fig. 11(b)).

Figure 11 indicates that most of the time, bimodal events' warm mode is very near the freezing point, irrespective of the time of year. In contrast, the cooler of the two modes varies greatly depending on when the forecast is initialized, with periodic behavior that aligns with seasonality. The lack of seasonal variability in the warmer mode that is very clearly evident in the cooler mode points towards a constant boundary forcing that exists in the warmer mode but not the cooler. The fact that the

warmer mode is almost always near the freezing point (what would be expected for the polar oceans) while the cooler mode is some temperature below freezing, indicates that the warmer mode is representative of a state with open ocean while the cooler mode is a sea ice state. In the former, the relatively warm ocean can flux heat to the overlying atmosphere, helping regulate the temperature irrespective of the atmospheric state (little spread in temperature PDF of Fig. 11(b)). In contrast, if sea ice is



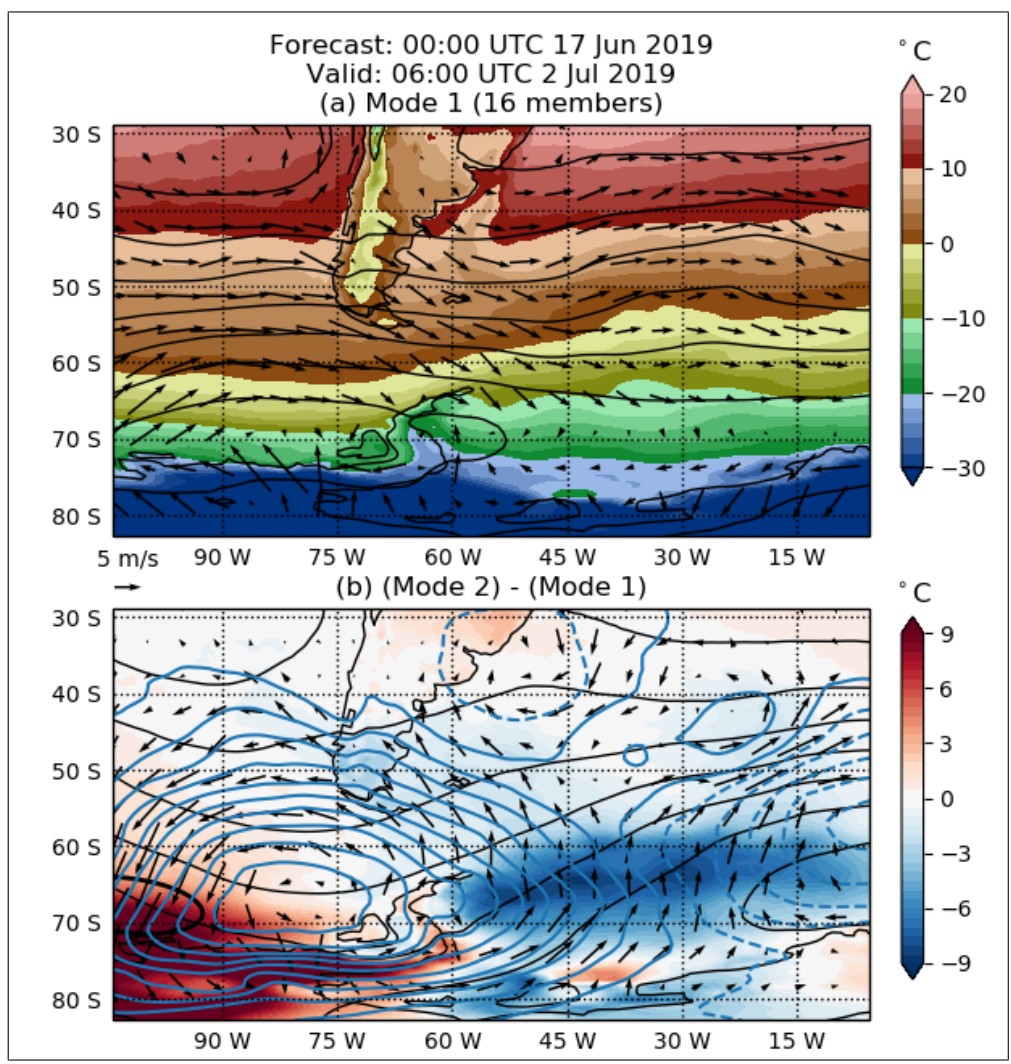

**Figure 10.** As with Fig. 7 but for a Southern Ocean forecast initialized on June 17, 2019 and valid for July 2, 2019.

present, there is no heat flux and 2-meter temperature will have greater variability based on day-to-day atmospheric conditions
(hence the greater spread in the PDF of Fig. 11(b)).

Using this as a guide, the bimodality within this region could be developing in two different ways: differing advective
influences, where flow from sea ice to ocean versus ocean to sea ice will have widely different effects; or, perhaps a dynamical
forcing on the sea ice field itself. In the latter case, for the particular example in Fig. 10, mode 2 has warm air advection from
the north which may push ice southward, exposing the relatively warm ocean, resulting in warmer 2-meter temperatures in the
western portion of the map. Similarly, cold air advection from the south may pull already developed ice sheets further north,
removing the temperature flux from ocean to atmosphere in these northerly locations; this would result in colder temperatures
in the eastern portion of the map. The separation of ice from Antarctica's coast could be one explanation as to why the cold




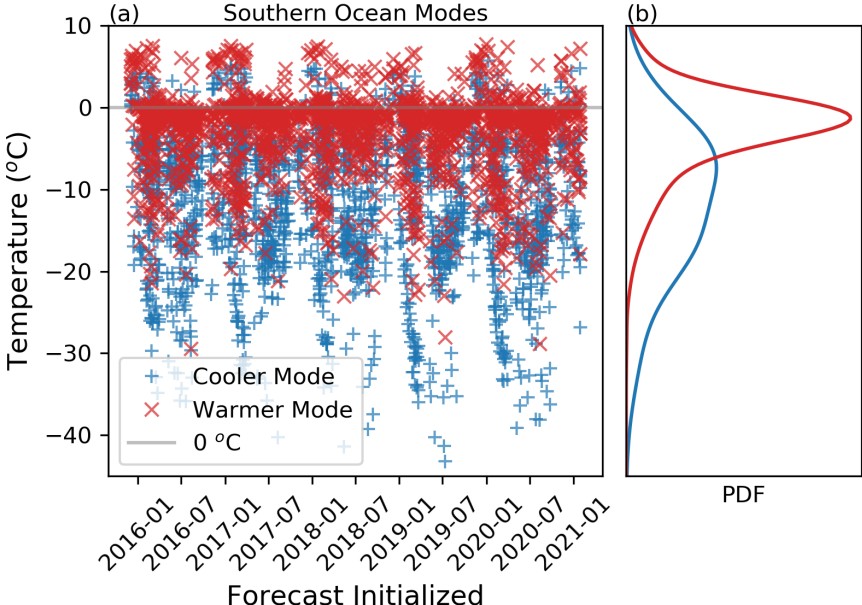

**Figure 11.** The most common temperatures of each mode for every bimodal event in the Southern Ocean region during MJJAS and NDJFM. Blue is representative of the most common temperature of the colder mode in the bimodal PDFs of Southern Ocean clusters, red ticks depict the warmer mode. A gray line has been marked for 0 $^o$C.

anomaly is further equatorward than the warm anomaly, as well as why a local warm anomaly is present just to the east of Antarctic peninsula where wind is divergent off the shore. However, the bimodality that develops may be a combination of
both the advective and dynamical influence.

This region is generally a very dynamically active portion of the world, where topographic forcings by the Antarctic peninsula produce unique and persistent wind field regimes that may explain some of the development in bimodality in this region (Schwerdtfeger, 1975; Massom et al., 2008; Hosking et al., 2013; Elvidge et al., 2016; Laffin et al., 2021). The dynamically forced sea ice states that are distinct between the two modes may be a result of an eastward propagating wave which com-
monly occurs throughout this region (Baba and Wakatsuchi, 2001). These waves are closely coupled with sea ice dynamics in this region and are found to have the greatest influence on sea ice formation in regions where meridional wind velocity is the greatest (Baba et al., 2006) (a pattern that is exhibited in Fig. 10(b)). Separately, the Antarctic Circumpolar Current acts as a sharp sea surface temperature (SST) gradient which, depending on direction of low levels winds, can produce very different near-surface temperature tendencies. This hypothesis may be supported by the distribution of bimodal forecast occurrences
exhibited in Fig. 3(b), which contains a maximum surrounding the entirety of the pole, mimicking that of the ACC; note that the ACC in part also controls the northward extent of the sea ice field (Martinson, 2012), which may coincide with the behavior exhibited in Fig. 11. Finally, aligning with dynamical activity within this region, there have been studies that have focused on





non-Gaussianity in the temperature field near the southern hemisphere storm track (Tamarin-Brodsky et al., 2019; Garfinkel and Harnik, 2017). These studies have identified warm skew south of the storm track axis (60 $^o$S) and cold skew to the north

of the storm track axis (38 $^o$S). These two regions align with maxima in bimodality as apparent in Fig. 3(b), with a minimum in between that may be representative of the center of the axis. The skew in these studies is identified to be partly as a result of Rossby wave breaking, events which may induce effects on the low level wind field. Rossby wave breaking events would be expected to occur to a greater extent during the cold season, when baroclinic forcings are greater (aligning with the occurrences in Fig. 3).

Thus, the primary processes leading to bimodal events within this region may shift seasonally. Perhaps in the cold season (Fig. 9(b)), events are moreso as a result of Rossby wave breaking induced low level wind field departures and their interaction with the sea ice field near the ACC. These Rossby waves would propagate west to east, without much zonal interaction from the continents, aligning with the event velocity data in Fig. 8(e). However, in the warm season, when the marginal ice zone is nearer that of the Antarctic continent and Rossby wave breaking frequency decreases, dynamical interaction of highs and lows

with the Antarctic peninsula may become the primary process contributing to bimodality. This could create more variable types of events due to the unique topography of Antarctica, as mentioned previously; this trait is perhaps reflected in the greater span of velocities that occur in Fig. 8(f) compared to Fig. 8(e). However, regardless of the process, most of the time, it is evident that these events are associated with a relatively warm near-freezing mode and a cooler below-freezing mode. Once again this methodology has indicated connections between bimodal events and well-documented common atmospheric phenomena that

occur within a region, a guiding motivation of this study.

### 3.3 North Atlantic Region

Figure 12 depicts histograms of event persistence, area and velocity for the North Atlantic region's cold and warm season. The median event persists for roughly 2 days, with slightly longer events occurring on average during the cold season. As seen in the previous two regions, the median event area approximately doubles in size for forecasts initialized during the cold season

versus during the warm season. Events propagate at a speed of 5-10 ms$^{-1}$, with slightly faster events generally occurring on average during the cold season. Roughly 2/3 of the time, events have an eastward direction of propagation in the cold season, with the remainder of the motion being predominantly south-eastward. However, the direction of propagation is more evenly split between eastward and south-eastward motion for warm season forecasts.

The centroid density and individual tracks for the warm and cold season for the North Atlantic region are plotted in Fig. 13.

During the summer season, events are concentrated between 70 $^o$W to 35 $^o$W and 35 $^o$N to 45 $^o$N, with a maxima of over 25 event days per season; this maximum is located near the Gulf Stream and thus the event dynamics may be linked to air-sea interactions over strong SST gradients. During the North Atlantic cold season, bimodal events appear to occur over a greater range of locations, with relatively even frequencies over much of the region depicted; however, there are notable maxima located near the Gulf of Mexico and the Gulf Stream. Reflective of what is depicted in Fig. 12, individual tracks appear to

mainly progress west to east, with some northwest to southeast propagation especially prevalent in the cold season's Gulf Stream maximum.





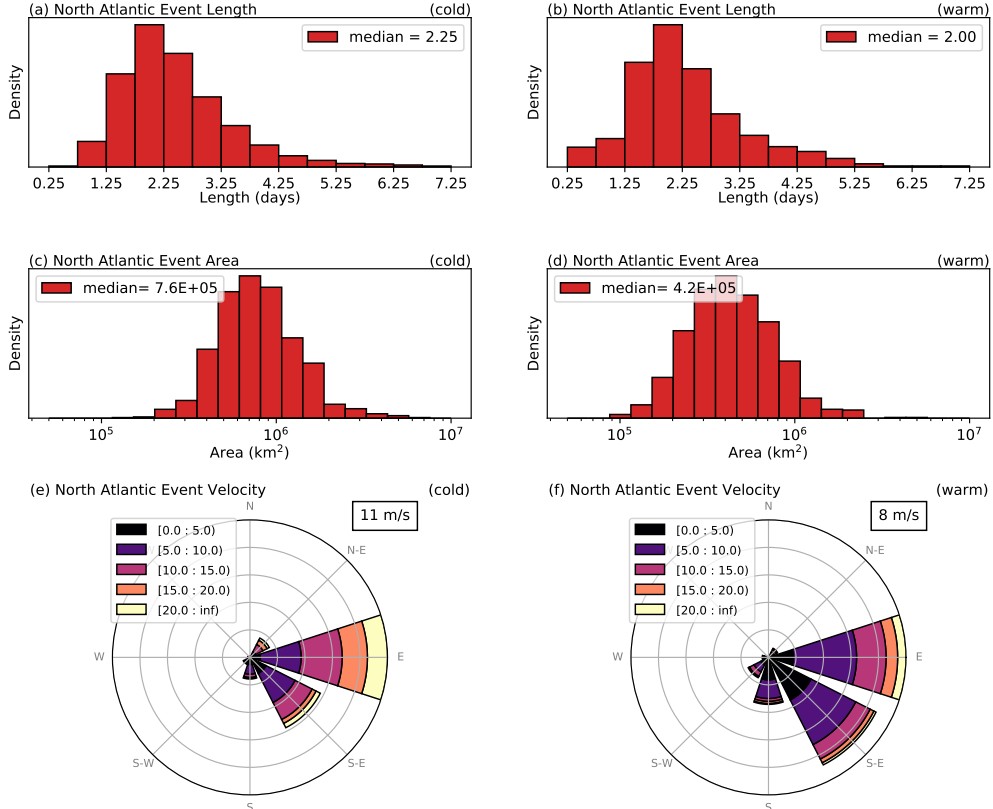

**Figure 12.** As with Fig. 5 but for the North Atlantic cold (November, December, January, February, March) and warm (May, June, July, August, September) season.

The evidence pointing towards any single phenomenon in this region is less clear than the South American case. The Atlantic maximum in Fig. 13 may indicate a forcing from sea surface temperature gradients near the Gulf Stream. This temperature gradient would be relatively consistent through both seasons, which mirrors the persistent maximum in centroid location.
However, other processes are clearly at play, especially in the cold season, causing a widely distributed frequency. The typical speed of the bimodal events as indicated by Fig. 12 is roughly 5-10 ms$^{-1}$, suggesting that they are not fixed relative to SST gradients and that atmospheric processes likely play a significant role in the evolution of these events.

In order to evaluate possible explanations, a specific case study is once again examined. The forecast depicted in Fig. 14 is initialized during the Northern Hemisphere cold season and is valid for March 6, 2018. This forecast's bimodality is associated
with very different geopotential height structures between the two modes. Mode 1 is characterized by a kinking in the isobars that looks similar to an equatorward anticyclonic Rossby wave breaking (LC1) event (Thorncroft et al., 1993). In contrast, mode 2 has a geopotential height structure that reflects a cutoff low. At this particular stage of the wave breaking event (mode 1), there is a large component of southward geostrophic flow around 55 $^o$W, advecting cold air towards the equator. This creates







**Figure 13.** As with Fig. 6 but for the North Atlantic cold (November, December, January, February, March) and warm (May, June, July, August, September) season.





comparatively a much warmer mode 2 which does not have this feature. In this case, the clustering process and ellipse fitting
does a relatively good job at capturing the location of maximum 2-meter temperature departure between the medoid's modes.

Wave breaking events are nonlinear and inherently hard to predict. It is not surprising that predicting when, where or to
what magnitude this process will occur would lead to spread in a forecast distribution (and thus perhaps bimodality). The high
rates of bimodality in this region would fit this explanation as this region also has a climatological maxima in Rossby wave
breaking events (Homeyer and Bowman, 2013), and while this is a different storm track region than those studies conducted
in the southern hemisphere which linked non-Gaussianity and Rossby wave breaking (Tamarin-Brodsky et al., 2019; Garfinkel
and Harnik, 2017), similar processes could be at play in this portion of the northern hemisphere storm track. Additionally,
since the Gulf Stream acts as a source of baroclinic instability, it allows for the forcing to persist during both seasons which
is consistent with the bimodality occurrences expressed in Fig. 3. Furthermore, the deepening of the trough and contortion of
the isobars that occurs as a wave passes over the Gulf Stream may partly explain the southeastward component to the event
velocity seen in Fig. 12(e) and Fig. 12(f).

Although the forecast depicted in Fig. 14 is representative of many of the cases analyzed, the distinct tracks that occur near
the Gulf of Mexico and within Canada in Fig. 13 indicates that multiple unique phenomena leading to bimodality may be
occurring within this region.

## 4 Application to Weather Regimes

The notion of bimodality evokes the possibility for the atmosphere to locally drift into notably different states. This raises the
question of a possible connection to weather regimes. Michelangeli et al. (1995) defines weather regimes by three criteria:
atmospheric states which exhibit recurrence, persistence, or are quasi-stationary. Recurrence implies an atmospheric state that
is relatively likely to occur (Kimoto and Ghil, 1993). Persistence implies that when the state occurs, there are some lasting
effects on a designated timescale. Finally, quasi-stationary implies that the time derivative of the large-scale dynamical forcing
approaches zero at some point (Toth, 1992). Should an atmospheric state express regime-like qualities and should its effect
on ensemble spread be consistent, model spread may be adjusted appropriately to improve forecasts. With the insight that has
been gained from the presented findings, the bimodality that develops may put within the context of 'regimes'.

One possible explanation for bimodality in forecasts may be part of the ensemble predicting one regime while the rest of the
ensemble predicts another, reflecting two metastable states as different modes in the forecast distribution. While others have
attempted to identify regimes through the use of clustering and principle component analysis (Corti et al., 1999; Smyth et al.,
1999; Straus and Molteni, 2004; Casola and Wallace, 2007), most of these use spatial or temporal correlations in the values of
the forecast fields themselves, such as the clustering of similar geopotential height fields, or have algorithms that preferentially
group forecasts near each other in time or space. In this study, however, the methodology presented is based entirely on the
structures of forecast distributions without any defined spatial or temporal dependence. Should bimodality be linked to regime
behavior, composites of bimodal forecast distributions may thus offer unique information about these regimes that could not
be discovered in traditional regime identification methods. For example, this methodology would present a new way in which







**Figure 14.** As with Fig. 7 but for a North Atlantic forecast initialized on February 26, 2016 and valid for March 6, 2018.



the 'recurrence' of regimes could be characterized using bimodal distributions, where an ensemble PDF serves as an estimate of the likelihood of the different atmospheric states. Furthermore, the identification of dynamical events linked to bimodality may give an indication of the processes that certain regimes are sensitive to. If two modes develop within a forecast distribution and these are connected to two separate regimes, the processes that are occurring at the time of the bifurcation may indicate processes that contribute to regime transitions.

It is immediately obvious that the hypothesized events leading to bimodality in this particular study are not large-scale weather regimes in the 'traditional' sense (such as the NAO$^+$, NAO$^-$, Atlantic Ridge, and Scandanavian Blocking regimes from Cassou (2008) and others). However, the recurring events that do seem to develop (i.e. cold air incursion events in South America or the very differing temperature modes in the Southern Ocean region) not only express similar qualities (recurrent and quasi-stationary) but also have important implications for humans and climate (Hamilton and Tarifa, 1978; Fortune and Kousky, 1983; Poveda et al., 2020; Schwerdtfeger, 1975; Massom and Stammerjohn, 2010). Furthermore, blocking events, which the bimodal events triggered by the Andes could be characterized as (and perhaps some of the events near the Antarctic peninsula), are well studied weather regimes (Schwerdtfeger, 1975; Charney and DeVore, 1979; Vautard, 1990; Garreaud, 2000; Orr et al., 2008; Domeisen et al., 2020). Since this study has focused only a few geographical regions with a methodology that could be further improved, we speculate that the methodology could be profitably adapted to identify more traditional weather regimes.

## 5  Conclusions

This paper introduces novel methodology based on ensemble distribution similarity to help study bimodality exhibited in subseasonal-to-seasonal 2-meter temperature forecasts from ECMWF, a topic introduced in Bertossa et al. (2021). While there are still pathways for optimization of the algorithm, the premise has proven useful in connecting bimodality to coherent atmospheric phenomena, a primary goal of this study. Bimodal clusters are found to have spatial and temporal coherence that can be identified by the mode membership of the ensemble alone. While being able to characterize events based on similarities in forecast behavior makes sense in theory, it is remarkable how well the idea behaves in the context of bimodality. Furthermore, the authors are unaware of such an approach being used previously to characterize weather events, despite the practicality of wanting to understand how specific forecast behavior can be linked to weather phenomena. The coherent groups of bimodal forecasts found with this methodology are deemed 'bimodal events'.

The general similarities in bimodal event length and speed between the three studied regions (South America, the Southern Ocean and the North Atlantic) may indicate a common forcing mechanism, such as the propagation of atmospheric waves. Furthermore each region's case study is characterized by differing geopotential height structures between the two modes (Fig. 7, Fig. 10 and Fig. 14), where one mode is typically more zonal than the other mode. However, each region's bimodal events appear to be unique based on large-scale dynamical interactions with local processes such as topography, SST gradients and the cryosphere.



While in general, events last 1-3 days and cover a spatial area on the order of $10^5$ to $10^6$ km$^2$, there are distinctions between events. One example is the Andes region, which has events that are relatively stationary with some south-to-north propagation versus the North Atlantic region which has events that propagate relatively continuously at 5-10 ms$^{-1}$ west-to-east.

The combination of the general behavior and case studies of bimodal events has allowed us to make inferences for several weather phenomena that may be leading to outbreaks in bimodality. In the South America region, the most common bimodal event appears to be due to one mode developing a blocking pattern which results in cold air incursions along the eastern flank of the Andes, while the other mode has flow that remains relatively zonal. In support of this finding, these two states represent the two most common modes of variability according to principle component analysis conducted in Compagnucci and Salles (1997). In the Southern Ocean, low level winds' interaction with sea ice appears to be the main cause of bimodality in forecasts. These winds may be attributed to blocking type regimes caused by the Antarctic peninsula, Rossby wave breaking that occurs near the marginal ice zone, or perhaps some other mechanism in this dynamically active region. Finally, in the North Atlantic, bimodal events appear to be associated with the development of baroclinic waves in the western Atlantic. This region has been identified as a common area for Rossby wave breaking, events often responsible for the transition between or the maintenance of an already existing regime (Michel and Rivière, 2011). This methodology may present a means to study such regime behavior, which is found to be important for the improvement of forecast skill (Allen et al., 2019), understanding energy balance (Yu et al., 2017) and even renewable energy (Garrido-Perez et al., 2020).

It is still unclear what triggers the separation between the two modes to create a bimodal event, rather than just large ensemble variance. Why ensemble members group together and then begin to diverge is indicative of distinct feedback mechanisms within each mode, but this process potentially varies from region to region or even from event to event. To answer these questions, more systematic studies or the use of conceptual models are most likely needed. However, features of the boundary conditions that favor the advection of distinctly different air masses come out as a frequent ingredient in the cases examined here: a north-south topographic barrier, or a sharp change in the boundary conditions such as sea-ice or a western boundary current. Thus, the use of conceptual models will likely need to contain some representation of this component. There is no indication in the literature that these hypothesized weather phenomena should have such an effect on the forecast distribution (i.e. lead to the development of two modes rather than just large variance), and thus a deeper understanding of the dynamics underlying the phenomena creating these bimodal outbreaks may be gained through these studies as well.

## Appendix A: Clustering Routine

Clusters of similar forecasts based on ensemble membership vectors are formed using a hierarchical clustering routine (Sasirekha and Baby, 2013), specifically Python's scipy.cluster.hierarchy.fclusterdata function (scipy v1.4.1) which performs hierarchical clustering using the single linkage algorithm with a custom distance formula.



The custom distance formula used to determine the similarity between two forecasts' membership vector $\boldsymbol{x}$ and $\boldsymbol{y}$ is defined as:

$$465 \quad d(\boldsymbol{x}, \boldsymbol{y}) = \frac{1}{\sqrt{C_x C_y + W_x W_y}} \sum_{i=1}^{N_E} (x_i - y_i)^2 \qquad \text{(A1)}$$

where $N_E$ is the number of members in the ensembles, $C_x = \sum_i^{N_E} x_i$ is the number of members in the cold mode of ensemble $\boldsymbol{x}$ and $W_x = N_E - C_x$ is the number of members in the warm mode of ensemble $\boldsymbol{x}$. $C_y$ and $W_y$ are defined similarly. The expression is symmetric, in the sense that $d(\boldsymbol{x}, \boldsymbol{y}) = d(\boldsymbol{y}, \boldsymbol{x})$. The normalization factor is largest when the cold modes of both forecasts are either much smaller or much larger than the warm modes ($C_x << W_x$ and $C_y << W_y$, or $C_x >> W_x$

470    and $C_y >> W_y$). Conversely it is smallest when the cold modes of each forecast are very different sizes ($C_x >> C_y$ and $W_y >> W_x$, or $C_x << C_y$ and $W_y << W_x$). It has been included to reduce the impact of a single member being in a different mode when one of the two modes is much larger than the other in both forecasts. Other normalization factors are also possible and several other expressions were also tested; the form (A1) is found to give the best results (generate the largest clusters with lower intra-cluster distances). An example of this process for two synthetic 5-member forecasts is depicted in Fig. 1(b).

475    What determines a cluster is an iterative routine involving a cluster center, or medoid, whose distance is minimized from surrounding points. The number of clusters can be predetermined either by a priori information or via a 'distance threshold', defined as $d_c$. $d_c$ specifies the maximum inter-cluster distance allowed, where smaller distances would imply more clusters; in this case, the clustering algorithm decides how many clusters will be present. As the threshold approaches 0, each point would be its own cluster. The value for $d_c$ used in this study is 1/37. This value was chosen through trial and error based on

480    maximizing cluster coherency and occupancy. This is a similar method to which the normalization factor for Eq. (A1) was tested. A more rigorous optimization of these two parameters may offer a better means of studying these events, however, is unnecessary for the exploratory nature of this manuscript.

*Data availability.* The ECMWF ENS extended forecasts are accessible through the ECMWF website (https://www.ecmwf.int/en/forecasts). ERA-Interim data can be found at http://www.ecmwf.int/research/era.

485    *Code availability.* The code for data analyses and plots is based on the free Python software. Scripts are available upon request.

*Author contributions.* CB conducted analysis and prepared text. PH supervised, aided in the analysis and contributed to the writing of the text. AD and RP aided in the development of research questions and hypotheses. All authors were involved in the revision of the text.





*Competing interests.* The authors declare that they have no conflict of interest.

*Acknowledgements.* We thank Cornell University for funding and the use of materials and facilities, as well as the European Centre for
490   Medium-Range Weather Forecasts (ECMWF) for providing the dataset used. The authors also acknowledge the data center ESPRI/IPSL for
their help in storing and accessing the data.



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
