# Peer review of "Bimodality in Ensemble Forecasts of 2-Meter Temperature: Event Aggregation"

_EGUsphere, 2022_

## Referee Comment (RC1)

Review of manuscript 2022 – 601 submitted to Weather and Climate Dynamics

**Bimodality in Ensemble Forecasts of 2-Meter Temperature: Event Aggregation**

by C. Bertossa et al.

This study investigates the emergence of bimodality in 2m temperature in subseasonal-to-seasonal forecasts of the ECMWF model. Building on a previous study that introduced a detection method for bimodality, this study introduces a clustering method that helps to identify "bimodal events" and study their characteristics (such as duration, spatial scale, propagation). This method is then applied to study bimodality in three regions: South America, the Southern Ocean, and the western North Atlantic. Case studies indicate that interaction of the large-scale atmospheric flow with boundary conditions (orography, sea ice, SST gradients) are a major cause of bimodality. I find that this approach can lead to interesting insights not only into why bimodality occurs in subseasonal prediction systems, but also into the general workings of the climate system. Accordingly, I find the paper fits well into the scope of WCD.

While I have no concerns about the methodology as such, I think that the exploration of the physical causes of bimodality remains a bit vague and  more could be done to pinpoint the actual causes of the bimodality. Some suggestions in that regard are given below.

Overall the paper is well written and figures are readable. However, the language is sometimes unclear and the method could be better explained. Also, there is a lot of jargon, which may be understood by specialists in subseasonal-to-seasonal prediction, but not the general readership of WCD. Since the study is certainly of interest for a broader readership in the climate dynamics community, I suggest the authors reduce the use of specific jargon and try to better explain certain concepts.

**General comments:**

1. What do you mean by the term "atmospheric events"? This term is used throughout the paper, but it remains unclear what exactly you mean by it. Do you think of synoptic events that might cause some ensemble members to follow a very different trajectory in phase space than most others, hence causing bimodality, or rather slower processes such as the continuous interaction of the atmosphere with the boundary conditions? Reading further ahead, I assume it is the latter. I think it is necessary that the term "atmospheric events" is precisely defined already in the introduction.

2. The geopotential height patterns are reminiscent of Rossby wave breaking events in this region causing cold surges in Brazil, see Sprenger et al. (2013). Hence, I was wondering whether the bimodality is related to the presence and absence of Rossby wave breaking? Considering Rossby wave breaking may give a more direct physical linkage to the causes of the bimodality.

*Sprenger, M., Martius, O. and Arnold, J. (2013), Cold surge episodes over southeastern Brazil – a potential vorticity perspective. Int. J. Climatol, 33: 2758-2767. https://doi.org/10.1002/joc.3618*

3. The relationship of bimodality in the Southern Ocean region to sea ice is interesting and appears plausible through the different heat capacities of water and ice, resulting in a damping effect on temperature variability in the former case.

   The explanation of why differences in sea ice state occur, however, remains overly vague. Generally, sea ice in this region reacts strongly to persistent wind anomalies, which can push the ice edge far away from its climatological position. Hence, It appears to me that it is not single synoptic events that cause the bimodality, but rather the accumulated effect of anomalous winds (for example through several cyclones passing through the Amundsen and Bellingshausen Seas) over one or two weeks, hence linking these events to longer timescales. Could the authors look into circulation anomalies in the preceding weeks?

4. Could explosive cyclogenesis play a role for the North Atlantic events? This region is well known for the frequent occurrence of bomb cyclones and Fig. 14b suggests the presence of a deep low in mode 2 which is absent in mode 1. Exploring this might give you a more direct physical linkage between the SST gradient associated with the Gulf Stream, which is known to play a crucial role for the rapid intensification of cyclones, and bimodality.

**Specific comments:**

L5: The phrasing here is a bit awkward. Understanding the origin of bimodality does not affect the skill of the forecasts but it helps understanding why the skill of forecasts sometimes is much worse than otherwise.

L29: Please specify what you mean by "dressing method".

L37ff: The data used in this study should be explained in the data and methods section, not the introduction.

L53ff: Consider moving this paragraph to the conclusion section. The introduction is meant to expose the open questions guiding the study based on the existing literature.

L71: Please specify what you consider the cold and warm modes of a forecast. I assume that these are the two modes of the bimodal distribution.

L71: And related to the above: how do you decide whether an ensemble member belongs to the cold or warm mode? The two single distributions constituting a bimodal distribution will normally overlap. How then do you attribute one member to a specific mode if it lies between the two?

L92: Please explain what the occupancy is and how the value of 32 follows from the previously said.

L164: What do you mean by erroneous area?

L155ff: If I understand correctly, you only use the ellipse but not the box. In that case there is no need to report on the definition of the box if you don't use it.

L222: Are you now considering the centroid or the ellipse center? As I understand the centroid

may be different from the center since not all points in the ellipse may exhibit bimodality.

L247: Why do you refer to modes 1 and 2 instead of cold and warm modes? Please be consistent with the nomenclature throughout the paper.

L249: I don't really see this wrapping of the high around the Andes. To me the high seems mostly confined to west of the Andes.

L279: Do you mean **north** of the Antarctic Peninsula?

L288: How is the temperature of a mode defined? Is the temperature taken as the mean over the ellipse of each event or for all individual grid points?

L303: I assume you are referring to the flow in the Bellingshausen and Amundsen Sea west of the Antarctic Peninsula. In the Weddell Sea the flow is northward.

L305: ice sheets is probably not the correct term for sea ice

L363:

L376: I don't understand the sentence "since the Gulf Stream acts as a source of baroclinic instability". How does baroclinic instability allow for the forcing to persist during both seasons? What do you mean by this?

**Figures:**

- Figure 1 is not well embedded in the text and I am not sure whether it is really needed. If you decide to keep it, it should specifically be used to illustrate the methodology, which is not the case right now.

**Technical corrections:**

L3: introduces **a** novel methodology

L24: bimodality **is** linked

L25: Consider merging this one-sentence paragraph with the next one.

L31: The sentence from "... noticable improvements..." onward does not seem to be grammaticaclly correct.

L42: guide **the** analysis

L86: Rephrase as "We then define the coherency of a cluster as the number ..." or similar.

L90: forecast **lead times** (also elsewhere, "leads" sounds overly sloppy)

L132: is found → is identified

L146: to **grow steadily**

L156f: **northernmost, southernmost etc.**

L158: is found → is identified / is determined

L180: exhibited in → shown in

L186: Please rephrase the sentence "Refer to left panel..."

L206: Question mark missing

L221: Rephrase as "Next it is explored where..."

L235: **is** identified

L237: **from** south to north

L238: **from** west to east

L249: The sentence "The result..." is essentially a repetition of what has just been said. Suggest to remove.

L265: Figure 7 → Fig. 7

L297: flux heat → give away heat

L326: delete as in "to be partly as a result of"

L357: than **in** the South American

L418: introduces **a** novel methodology

Figures: References to individual panels of a figure should be: Fig. Xa not Fig. X(a). Multiple panels should be references as Figs. Xa, b not Fig. Xa and Fig. Xb.

Caption Fig. 2:
- **The** thick black **line** indicates…
- Instead of stating that the red dashed and the solid blue line with white outline are explained in the text, simply say that they are selected members discussed in the text.

Caption Fig. 8 and many others. "As with Fig. X" should be "As Fig. X"

---

## Referee Comment (RC2)

Review of the manuscript:

Bimodality in Ensemble Forecasts of 2-Meter Temperature: Event Aggregation

by Bertossa et al.

This manuscript addresses an interesting topic. Based on the comments below, I recommend publication subject to major revisions.

Major comments:

1       The methodology used in this study is perhaps unnecessarily complicated and uses lots of arbitrary thresholds and other subjective elements. This undermines the robustness of the study. This is beyond something that is not elegant – it is cumbersome and hard to defend. I recommend that the Authors scrutinize their methodology and eliminate unnecessary conditions or base their choices on more objective conditions. Examples include: l.111, 116-117, 128, 136-137, 154 (subjective choice of 3 regions), 160, 166, 179-185. Some of the arbitrary choices are necessitated by earlier subjective conditions etc. Several of these choices appear to be made so a desired outcome is achieved (e.g., l.160, 179-185), undermining the generality of some of the findings. For further details, see detailed comments below.

2       The study presents results suggesting bimodality in the distribution of ensemble members. This condition, however, is never established objectively. I recommend that the Authors consider developing an algorithm to test whether the results they present are statistically (in)distinguishable from surrogate data generated using a null hypothesis that ensemble forecasts are normally distributed. Are results over the entire sample (i.e., over selectively chosen data) indistinguishable from randomly generated samples from a normal distribution? Without such a test, the results are not convincing and would be appropriate to present only as a short note.

3       In my view, no major conclusion appears to emerge from Section 3. The discussion is mostly descriptive, with no insight into the dynamics leading to the emergence of bimodality. As the results in Section 3 may be limited in significance, I find the number of figures (10) excessive. I recommend that the Authors significantly reduce the number of figures and resort to using short verbal statements to describe less important results. Also, as some of the results for the three regions studied are similar, results for the three regions could be compared and discussed together, shortening the text.

4       The Discussion in Section 4 is interesting but appears somewhat preliminary. A more logical organization of the material and a shorter presentation would improve this section.

5       Apart from its last paragraph, material in the Conclusions section also appears to be disorganized and premature. As the manuscript mostly presents hypotheses without objective validation, unless more evidence is added, some of the statements (e.g., l.420-421 should be tempered accordingly. Depending on how the Authors respond to the reviews, this section also needs a major revision.

6       The quality of the presentation is mixed. There are some parts in the Introduction that are well written. The bulk of the manuscript, however, needs significant improvements before possible publication. See detailed comments below for some examples. Some suggestions:
a) Clearly define all terms used in the study. Examples include: size, occupancy, points, forecast in the description of the methodology. Some terms may be poorly named, using commonly used words with a different meaning. In such a case, clear definitions are especially important.
b) Figures should be defined in their captions but the discussion of the results should appear in the text.
c) Replace hypothetical examples used in some figures (e.g., Fig 2) with examples from the actual data.

Detailed comments:
1) L. 25-26: "since the growth of perturbation variance in an ensemble"
2) Sentence in l. 29-30: this may need some additional explanation and/or references
3) L.34 – not clear
4) L.35 – "dressing methods" – please explain (or provide proper references)
5) L.42: "guide our analysis: (1) Is bimodality…"
6) L.47 – not clear
7) L.48, l.67, "ensemble spread" is customarily used to refer to the square root of the variance of ensemble members around their mean. I suggest you use "the distribution of ensemble members" or "ensemble distribution" when referring to specific behavior of such distributions, like bimodality
8) L.49: "answer question (1)"
9) L.51, "contribute to answering question (2)"
10) L.52. "naturally addresses question (3)"
11) L.53: Clusters need to be defined before they are discussed any further

12) L.65 – "concludes the study"

13) L.75 - how "groups of forecasts" differ from clusters?

14) L. 78-79 – Why is it so?

15) L.80 - Please clarify "based on their 2m temperature values"

16) Section 2.1 – The text is hard to follow, which prevents the reader to assess the material presented here.

17) L.92, end of sentence – not clear

18) L.111 – "few dozen forecasts" – why is this a problem?

19) L.128 - is this redundant, given a similar earlier statement?

20) L.140-141: not clear why this may be the case? This may be suggestive of a causal link, but may not necessarily be an indicator that full life cycles are assessed?

21) L.148 – Fig. 3?

22) L.150, "developing around true dynamical events" - an alternative explanation that needs to be evaluated by statistical tests is whether bimodality may emerge in normally distributed 2m temperature data due to sampling fluctuations, see major comment 2.

23) L.175-178: Admittedly, the results indicate spatiotemporal consistency. Due to dynamically conditioned covariance in meteorological data, this, however, would also appear if bimodality is a result of pure statistical sampling fluctuations in normally distributed data ("normal distribution hypothesis"). Can you substantiate your claim of dynamical origin for the emergence of bimodal signals by refuting a normal distribution hypothesis via specially designed tests (see major comment 2)?

24) L.198-199: perhaps not "optimizing" by refining or adding arbitrary choices, but rather revising the methodology may be needed to reach more firm conclusions?

25) L.204 – "lead to the emergence of bimodality"

26) L.238 – "eastward propagation"

27) L.240, Fig. 7 and similar figures: this is a rather unusual presentation of the data. The 2 panels use different formats, making a straight comparison difficult. Would it be more informative to show anomalous conditions (from the mean of the ensemble) in the same format for both modes?

28) L.242, "medoid" – not clear what this refers to

29) L.269, "exhibiting usefulness" - Admittedly, one of the two modes is found to be associated with a synoptic development documented earlier. In what sense does this make the methodology useful?

30) L.301-310: these are two interesting hypotheses, just as topographic forcing and Rossby-wave breaking ideas introduced later. The four hypotheses presented appear reasonable, and beg for validation. I wonder if the Authors

could experimentally confirm any of these hypotheses by examining sea ice and other relevant data? The discussion in this subsection is relatively long and most of it is speculative. Without such an analysis these are not more than interesting suggestions. I would like to suggest that the authors shorten this subsection, unless they wish to present evidence confirming or denying each hypothesis presented. As an example, l.369-370 could be removed without loss of general information.

31) L.354, " Reflective of what is depicted in Fig. 12" – superfluous, consider deleting
32) L.357, "in the South American region"
33) L.361, "The typical speed of the bimodal events as indicated by Fig. 12 is roughly 5-10 ms$^{-1}$," - redundant, given similar statement above; consider consolidating the two statements about speed.
34) L.382-383 – interesting hypotheses mentioned here, though without validation, they have limited value.
35) L.391, "model spread may be adjusted appropriately to improve forecasts" – please clarify
36) L.394, "metastable" – is this a good choice of word here?
37) L.410-411: "recurrence and quasi-stationarity"
38) L.422-426 - I find this confusing. Bimodality is a characteristic of forecasts, not weather events, isn't that true? Please reconsider.
39) L.428 - Is this contradictory with the discussion of multiple causes of bimodal forecasts in Section 3?
40) L.440: Has this been presented in Section 3? It appears to belong there.
41) L.445: "as an area of frequent Rossby…"
42) L.447-448 - this sounds a bit far-fetched and not supported by this study. Consider removing or qualifying this statement.
43) L.449 - Yes, more insight into this basic question would certainly make the manuscript more substantive.

---

## Author Comment (AC1)

The authors thank the reviewers for their feedback, we feel it has improved the quality of the manuscript. We have provided inline responses to each reviewer comment, as well as a reference to where such changes can be found in the revised text.

**RC1:**

While I have no concerns about the methodology as such, I think that the exploration of the physical causes of bimodality remains a bit vague and more could be done to pinpoint the actual causes of the bimodality. Some suggestions in that regard are given below. Overall the paper is well written and figures are readable. However, the language is sometimes unclear and the method could be better explained. Also, there is a lot of jargon, which may be understood by specialists in subseasonal-to-seasonal prediction, but not the general readership of WCD. Since the study is certainly of interest for a broader readership in the climate dynamics community, I suggest the authors reduce the use of specific jargon and try to better explain certain concepts.

We have made substantial efforts to reduce the jargon and have tried to be more explicit in defining the jargon that is still present. Furthermore, we more concisely and explicitly present specific definitions that we introduce for the clustering methodology/ tracking algorithm (Section 2).

We have also added several aspects of our analyses to better identify the physical causes of bimodality, though see our response to the first comment of reviewer 2 where we discuss the scope of our efforts.

General comments: 1. What do you mean by the term "atmospheric events"? This term is used throughout the paper, but it remains unclear what exactly you mean by it. Do you think of synoptic events that might cause some ensemble members to follow a very different trajectory in phase space than most others, hence causing bimodality, or rather slower processes such as the continuous interaction of the atmosphere with the boundary conditions? Reading further ahead, I assume it is the latter. I think it is necessary that the term "atmospheric events" is precisely defined already in the introduction.

In the revised text we remove the term 'atmospheric events'. This was originally meant to be a vague term to represent atmospheric phenomena that may lead to bimodal events without necessarily knowing what form these will take. However, we have rewritten the manuscript such that we introduce the term 'bimodal events' earlier and make it clear that we are trying to understand if (and why) they exist (Introduction).

2. The geopotential height patterns are reminiscent of Rossby wave breaking events in this region causing cold surges in Brazil, see Sprenger et al. (2013). Hence, I was wondering whether the bimodality is related to the presence and absence of Rossby wave breaking? Considering Rossby wave breaking may give a more direct physical linkage to the causes of the bimodality. Sprenger, M., Martius, O. and Arnold, J. (2013), Cold surge episodes over southeastern Brazil – a potential vorticity perspective. Int. J. Climatol, 33: 2758-2767. https://doi.org/10.1002/joc.3618

This is a very compelling argument, however, beyond a much more extended analysis, which is beyond the scope of this manuscript, it is difficult to confirm this hypothesis. In the revised manuscript, we have mentioned this as a possibility and include the reference (Section 3).

3. The relationship of bimodality in the Southern Ocean region to sea ice is interesting and appears plausible through the different heat capacities of water and ice, resulting in a damping effect on temperature variability in the former case. The explanation of why differences in sea ice state occur, however, remains overly vague. Generally, sea ice in this region reacts strongly to persistent wind anomalies, which can push the ice edge far away from its climatological position. Hence, It appears to me that it is not single synoptic events that cause the bimodality, but rather the accumulated effect of anomalous winds (for example through several cyclones passing through the Amundsen and Bellingshausen Seas) over one or two weeks, hence linking these events to longer timescales. Could the authors look into circulation anomalies in the preceding weeks?

This is a good point and the fact that the mean persistence of these events is only a couple of days to begin with, may indicate that dynamic (or thermodynamic) influences from the differing sea ice fields is not the main cause. We address this by looking at the sea ice fields of the two modes of several case studies to examine if they're identical or not. We find that while the sea ice fields do differ between the two modes (and bimodality develops along this difference), bimodality also exists away from this sea ice edge. It seems bimodality may in part be caused simply by the direction of low level winds over the sea-ice-to-ocean transition. We provide a more extended discussion of these processes in Section 3 of the revised manuscript.

4. Could explosive cyclogenesis play a role for the North Atlantic events? This region is well known for the frequent occurrence of bomb cyclones and Fig. 14b suggests the presence of a deep low in mode 2 which is absent in mode 1. Exploring this might give you a more direct physical linkage between the SST gradient associated with the Gulf Stream, which is known to play a crucial role for the rapid intensification of cyclones, and bimodality.

Similar to the Andes region, while this is a possibility it is difficult to state this for certain without a much more extended analysis. In the case studies that we examined in detail, explosive cyclogenesis does not seem to be at least the main driving factor, however, these events may certainly lead to some of the developed bimodality. We include this process as a possibility in the revised manuscript (Section 3).

**RC2:**

Major comments: 1. The methodology used in this study is perhaps unnecessarily complicated and uses lots of arbitrary thresholds and other subjective elements. This undermines the robustness of the study. This is beyond something that is not elegant – it is cumbersome and hard to defend. I recommend that the Authors scrutinize their methodology and eliminate unnecessary conditions or base their choices on more objective conditions. Examples include: l.111, 116-117, 128, 136- 137, 154 (subjective choice of 3 regions), 160, 166, 179-185. Some of the arbitrary choices are necessitated by earlier subjective conditions etc. Several of these choices appear to be made so a desired outcome is achieved (e.g., l.160, 179- 185), undermining the generality of some of the findings. For further details, see detailed comments below.

We thank the reviewer for their critical and constructive comments. We have substantially rewritten the manuscript in an effort to clarify the methodology, the scope of our study, and the main results.

We agree that our methods suffer from systematic selection biases—specifically towards larger events. However, the point of this methodology is not to accurately capture all bimodal events, but rather to demonstrate that they exist in the first place and to begin to connect them to several large-scale phenomena as a basis for future study. To carry out a systematic study of all bimodality present in these forecasts would be far too large a task for a single paper. We have reconstructed some of the text to try to more explicitly state this goal (Introduction and Section 2). A follow up study would be necessary to fully characterize, for example, if or what mesoscale events may lead to bimodality. Furthermore, we have removed some of the filtering that we perform on the clusters to streamline the methodology (with minimal change to results).

2. The study presents results suggesting bimodality in the distribution of ensemble members. This condition, however, is never established objectively. I recommend that the Authors consider developing an algorithm to test whether the results they present are statistically (in)distinguishable from surrogate data generated using a null hypothesis that ensemble forecasts are normally distributed. Are results over the entire sample (i.e., over selectively chosen data) indistinguishable from randomly generated samples from a normal distribution? Without such a test, the results are not convincing and would be appropriate to present only as a short note.

We have already performed such an analysis in the initial study of bimodality (see Bertossa et al. 2021). Bertossa, Cameron, et al. "Bimodality in ensemble forecasts of 2 m temperature: identification." *Weather and Climate Dynamics* 2.4 (2021): 1209-1224.

3. In my view, no major conclusion appears to emerge from Section 3. The discussion is mostly descriptive, with no insight into the dynamics leading to the emergence of bimodality. As the results in Section 3 may be limited in significance, I find the number of figures (10) excessive. I recommend that the Authors significantly reduce the number of figures and resort to using short verbal statements to describe less important results. Also, as some of the results for the three regions studied are similar, results for the three regions could be compared and discussed together, shortening the text.

We feel that the combination of the summarizing statistics and the case studies are necessary to explore the properties of events that lead to bimodality. This is a completely unexplored topic and so we intend to present a broad overview of each region. However, the additional analysis of the sea ice field, for example, has allowed us to refine our insights and shorten the text. Furthermore, we have shortened discussion within each region and added a general 'discussion' section in which similarities between regions are discussed, removing redundancies.

4. The Discussion in Section 4 is interesting but appears somewhat preliminary. A more logical organization of the material and a shorter presentation would improve this section.

We agree that with the present findings, much of this is speculative. We have removed this section and only place a shortened version in the conclusions section.

5. Apart from its last paragraph, material in the Conclusions section also appears to be disorganized and premature. As the manuscript mostly presents hypotheses without objective validation, unless more evidence is added, some of the statements (e.g., l.420-421 should be tempered accordingly. Depending on how the Authors respond to the reviews, this section also needs a major revision.

We have revised the conclusion section significantly and feel that with the new analysis, our results are significant and better supported by the analysis presented. However, we also reemphasize that the objective of our manuscript is more to identify these bimodal events rather than exhaustively characterize their behavior.

6. The quality of the presentation is mixed. There are some parts in the Introduction that are well written. The bulk of the manuscript, however, needs significant improvements before possible publication. See detailed comments below for some examples. Some suggestions: a) Clearly define all terms used in the study. Examples include: size, occupancy, points, forecast in the description of the methodology. Some terms may be poorly named, using commonly used words with a different meaning. In such a case, clear definitions are especially important. b) Figures should be defined in their captions but the discussion of the results should appear in the text. c) Replace hypothetical examples used in some figures (e.g., Fig 2) with examples from the actual data.

In the revised text, we more concisely and explicitly define terms in the methodology section and limit defining figures to just their captions rather than in-text. The authors feel that c) is important since it is intended to give a clear example of the clustering process and definitions. The actual data is less idealized and harder to present in such a simple manner.